# CVE-Factory: Scaling Expert-Level Agentic Tasks for Code Security Vulnerability

**Xianzhen Luo** [1][*] **Jingyuan Zhang** [2][*] **Shiqi Zhou** [1][*] **Jinyang Huang** [3][*] **Chuan Xiao** [1] **Qingfu Zhu** [1][†]
**Zhiyuan Ma** [4] **Xing Yue** [2] **Yang Yue** [2][†] **Wencong Zeng** [2] **Wanxiang Che** [1][†]

## Abstract

Evaluating and improving the security capabilities of code agents requires high-quality, executable vulnerability tasks. However, existing works rely on costly, unscalable manual reproduction and suffer from outdated data distributions. To address these, we present CVE-Factory, the first multi-agent framework to achieve expert-level quality in automatically transforming sparse CVE metadata into fully executable agentic tasks. Cross-validation against human expert reproductions shows that CVE-Factory achieves 95% solution correctness and 96% environment fidelity, confirming its expert-level quality. It is also evaluated on the latest realistic vulnerabilities and achieves a 66.2% verified success. This automation enables two downstream contributions. First, we construct LiveCVEBench, a continuously updated benchmark of 190 tasks spanning 14 languages and 153 repositories that captures emerging threats including AI-tooling vulnerabilities. Second, we synthesize over 1,000 executable training environments, the first large-scale scaling of agentic tasks in code security. Fine-tuned Qwen3-32B improves from 5.3% to 35.8% on LiveCVEBench, surpassing Claude 4.5 Sonnet, with gains generalizing to Terminal Bench (12.5% to 31.3%). We open-source all code, data, and models at https://github.com/livecvebench/CVE-Factory.

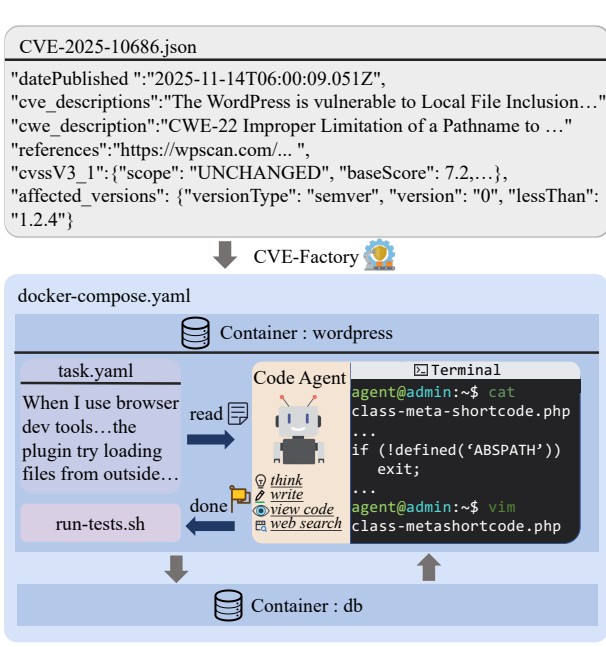

*Figure 1.* Comparison between raw CVE metadata and a comprehensive agentic task. (Top) Sparse CVE metadata consisting of vulnerability descriptions, classifications, and reference URLs. (Bottom) An agentic task including natural-language instructions, an interactive environment, and verification tests.

## 1. Introduction

AI-driven software development has enabled code agents to handle high-privilege tasks such as managing complex environments, executing scripts, and performing production-level deployments (Yang et al., 2025d). As these agents become widespread, the volume of AI-generated code grows explosively while human oversight diminishes proportionally, making security a critical requirement alongside functional correctness (Chen et al., 2025). Agents lacking sufficient security reasoning pose massive systemic risks: high autonomy combined with large-scale code output can propagate vulnerabilities at unprecedented speed (Su et al., 2025). Therefore, evaluating and enhancing the security proficiency of code agents has become an urgent challenge.

---

[*]Equal contribution [1]Harbin Institute of Technology (HIT) [2]Kuaishou Technology [3]Central South University (CSU) [4]University of Science and Technology of China (UTSC). Correspondence to: Qingfu Zhu <qfzhu@ir.hit.edu.cn>, Yang Yue <yueyang07@kuaishou.com>, Wanxiang Che <car@ir.hit.edu.cn>.

*Proceedings of the 43rd International Conference on Machine Learning*, Seoul, South Korea. PMLR 306, 2026. Copyright 2026 by the author(s).

A promising approach is to train and evaluate agents on realistic vulnerability repair tasks. Beyond static code snippets, such tasks require executable environments which agents can navigate codebases, execute commands, and iteratively refine solutions based on feedback. A complete task package must therefore provide a task description, an environment that faithfully reproduces vulnerabilities, and a suite of verification tests and solutions. However, existing task generation efforts fall short. While the community maintains CVELists [1], which document extensive collections of Common Vulnerabilities and Exposures (CVE), these sources provide only sparse descriptions and references, as shown in Figure 1. Prior works (Wei et al., 2025; Zhu et al., 2025; Wang et al., 2025) manually reproduce CVE metadata into tasks but this human-intensive approach fails to scale. CVEs involve heterogeneous programming languages and complex, unconfigured systems, costs experts on average 10+ hours per CVE (Mu et al., 2018). For instance, the WordPress setup in Figure 1 requires installing specific packages in a Dockerfile and configuring a backend MySQL database. Automated task generation frameworks from software engineering offer partial solutions but remain limited to Python or depend on well-configured repositories (Pan et al., 2024; Zhang et al., 2025b; Zeng et al., 2025; Yang et al., 2025c). The prior attempt at multi-agent CVE reproduction, CVE-Genie (Ullah et al., 2025), produces task formats incompatible with agent training.

To address these limitations, we present CVE-Factory, a multi-agent framework that automatically transforms CVE metadata into fully executable agentic tasks. A single agent is often intractable because CVE reproduction is an exceptionally long-horizon mission often exceeding 200k tokens (Zhang et al., 2025a).. We resolve this by decoupling it into three independent generation stages and three progressive verification stages, each handled by a specialized agent with focused context. Crucially, this isolation is balanced by a feedback mechanism that intelligently routes problems back to the responsible agent. Rather than constraining agents to fixed workflows, we grant them full autonomy within safety boundaries, allowing creative exploration while ensuring rigor through objective script-based verification. A central Orchestrator governs this entire lifecycle, managing agent activation, results validation, and the feedback loop. To enhance realism, task descriptions are formulated as user reports rather than technical CVE descriptions and require holistic validation of the entire system rather than isolated submodules.

We first evaluate the reproduction quality of CVE-Factory by reconstructing 215 expert-built tasks from PatchEval (Wei et al., 2025) using identical initial metadata. In cross-execution experiments, CVE-Factory solutions achieve a 95% pass rate on expert environments, while 96% expert solutions match ours. Furthermore, 74% of our tests are evaluated as equivalent or superior to expert versions. These results demonstrate that CVE-Factory achieves expert-level reproduction capability. We then evaluate real-world applicability by reproducing 454 recent CVEs (May–December 2025), with 66.2% passing rigorous manual validation. This process reveals an increasing proportion of vulnerabilities in AI-tools such as PyTorch. This insight motivates LiveCVEBench, a continuously updated benchmark of 190 tasks spanning 14 languages and 153 repositories that tracks the evolving threat landscape. With quality validated, we attempt the first large-scale synthesis of code security tasks, producing over 1,000 executable training tasks. Fine-tuning Qwen3-32B on distilled trajectories yields a 6.8× improvement on LiveCVEBench and 4.2× on PatchEval. Improvements also generalize to Terminal Bench (2.3×) (Team, 2025), confirming utility beyond security tasks.

Our contributions are summarized below:

- CVE-Factory: A multi-agent framework that autonomously transforms sparse CVE metadata into fully executable agentic tasks with expert-level quality.

- LiveCVEBench: A continuously updated benchmark of 190 tasks spanning 14 languages and 153 repositories tracking real-world distribution shifts.

- Scaling and Training: The first large-scale synthesis of code security tasks, producing over 1,000 executable environments. Fine-tuning Qwen3-32B yields 6.8×, 4.2×, and 2.3× improvements on LiveCVEBench, PatchEval, and Terminal-Bench.

## 2. Related Work

**Code Security** The primary task in code security is to locate and fix vulnerabilities while preserving functionality (Sanvito et al., 2025). The field has evolved through three stages. Early static approaches constructed ⟨vulnerable code, fixed code⟩ pairs from CVE commits for training (Fan et al., 2020; Fu et al., 2022; Ding et al., 2025; Steenhoek et al., 2025; Yang et al., 2025b; Simoni et al., 2025; Gao et al., 2024). Evaluation relied on static matching or manual review (Bhandari et al., 2021; Wang et al., 2021; So & Oh, 2023; Mou et al., 2025; Liu et al., 2025). As code interpreters became widely adopted, evaluation shifted to dynamic test execution (Gao et al., 2021; Bui et al., 2022; Wu et al., 2023; Hu et al., 2025). The rise of Code Agents has since transformed the paradigm toward autonomous exploration and repair within complete environments (Zhu et al., 2025; Yu et al., 2025; Wei et al., 2025; Mei et al., 2024; Lee et al., 2025; Zhang et al., b;a). However, these tasks still require manual construction (Zhu et al., 2025; Mu

[1]https://github.com/CVEProject/cvelistV5

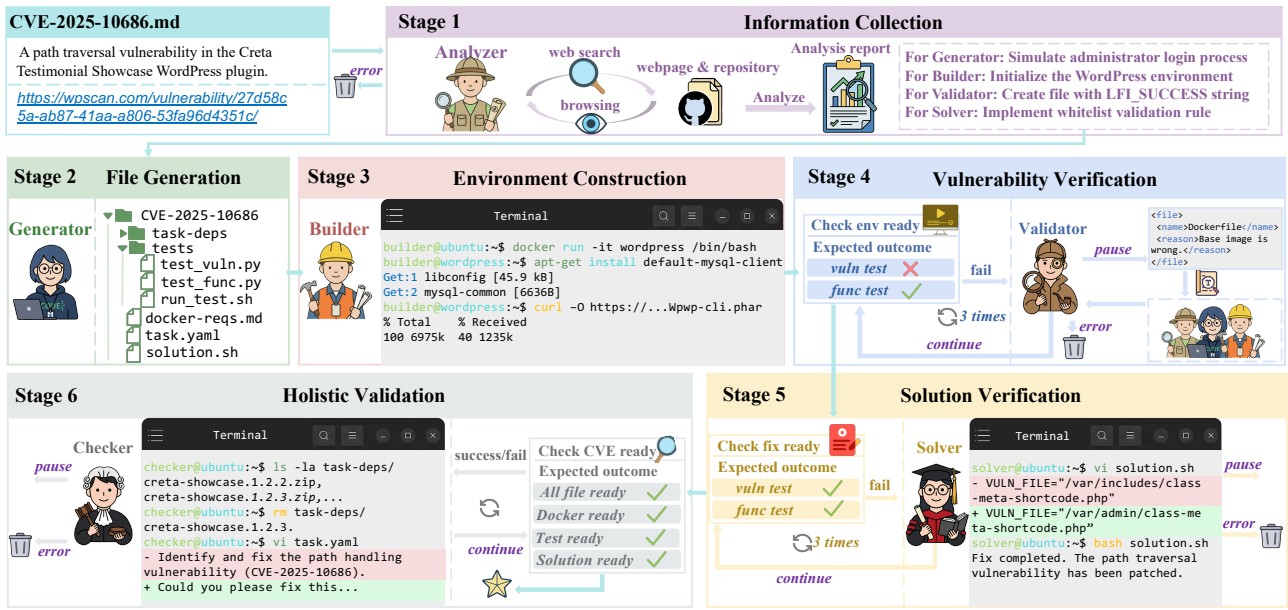

*Figure 2.* Overview of CVE-Factory. CVE Metadata are processed through six stages: three decoupling stages (Stages 1–3) generate task components independently, and three coupling stages (Stages 4–6) progressively verify and align them. A central Orchestrator manages the workflow, activating specialized agents and executing verification scripts per stage. Agents communicate with Orchestrator via `continue`, `error`, or `pause` signals, where `pause` triggers the feedback mechanism to route revisions to original file creators.

et al., 2018). This scarcity severely constrains agent-based evaluation and training at scale for code security.

**Agentic Task Construction** Automatically constructing executable tasks from static information is an active research area. SWE-smith and R2E-Gym (Yang et al., 2025c; Jain et al., 2025) use agents to explore environments but still require manual Dockerfile finalization. SWE-bench-Live and SWA (Zhang et al., 2025b; Vergopoulos et al., 2025) achieve full automation but only for Python repositories with standard metadata like requirements.txt. Multi-language frameworks such as SWE-Factory (Guo et al., 2025) remain limited in scale, covering only a dozen repositories. These methods are insufficient for CVEs, which involve diverse languages, complex service architectures, and incomplete environmental information. CVE-Genie (Ullah et al., 2025) made initial attempts at multi-agent CVE reproduction, demonstrating feasibility but without generating complete task packages including Dockerfiles, test scripts, and task descriptions. CVE-Factory addresses these gaps by automating CVE transformation at scale.

## 3. CVE-Factory

CVE-Factory is a multi-agent framework that transforms raw CVE metadata into verified, executable task packages. As illustrated in Figure 2, package follow the Terminal Bench (Team, 2025) and includes: `task.yaml` for instruc-

tions; `Dockerfile` and `docker-compose.yaml` for environment setup; `solution.sh` as the reference fix; and `run-tests.sh` to execute evaluation. For security tasks, testing is split into `test_func.py` (functional stability) and `test_vuln.py` (vulnerability presence before fixing and resolution afterward).

### 3.1. Architecture Design

**Task Decouple & Couple** Directly employing existing code agent frameworks like Claude Code to reproduce a CVE often fails. This mission requires transforming sparse descriptions into a complete environment, test suite, and solution. The high volume of required files and the long verification cycles typically overwhelm the agents. We address this by splitting the process into six stages: three decoupling stages (Information Collection, File Generation, and Environment Construction) that break the mission into independent tasks, and three coupling stages that progressively align these components to restore the original holistic task. Stage 4 aligns the environment with the tests to verify the vulnerability trigger, while Stage 5 aligns the solution with the verified environment to fix the vulnerability. By the time Stage 6 performs the final end-to-end verification, the task difficulty is significantly reduced because the core components are already synchronized. With this sequence of isolated generation and incremental alignment, our design ensures that the cognitive burden remains low while successfully restoring the full-scale task.

**Context Isolation & Reuse**  Each stage is assigned a specialized agent with an independent context. Agents do not share dialogue histories. Instead, essential knowledge is transferred via distilled Markdown files. This isolation prevents irrelevant information from consuming limited context windows. For instance, extensive Docker logs are largely irrelevant to subsequent verification. However, we also implement a feedback mechanism for scenarios where reusing previous information improves efficiency. When an agent identifies a fundamental flaw requiring file reconstruction, the system routes the failure back to the original creator. This allows the original agent to leverage exploration history for efficient repair rather than restarting from scratch.

**Agent Autonomy & Control**  Our agents are not restricted to predefined workflows or tools. Instead, each agent is a Claude Code session with the liberty to explore the workspace, execute commands, and adapt to feedback autonomously. However, such high autonomy requires constraints to prevent agents from being misled by prior information, executing unsafe operations, or making subjective judgments. First, we maintain information asymmetry by blinding Builder to pre-generated tests or solutions. This prevents agents from mocking the expected results and stops error propagation. Second, agents are prohibited from reading or writing files outside the designated working directory and cannot execute dangerous system commands. Finally, task completion is determined by objective static scripts rather than an agent's self-assessment.

**Orchestrator**  A central Orchestrator manages the entire reproduction process. It controls CVE state flow, executes verification scripts, and activating agents. Agents interact with it through a structured `agent-res.xml`, which has three signals. `continue` indicates that the agent considers its task complete. Orchestrator then performs static script validation, like ensuring required files are generated. If this validation fails, the error detail is sent back to the agent for further refinement. `Error` means that the CVE is determined as irreproducible by agent, prompting the Orchestrator to terminate the process. Finally, `pause` triggers the feedback mechanism. In this case, the agent identifies the problematic file and the reason for the revision. Orchestrator maintains a file ownership map to route revision requests to original creators. Once the creator resolves the issue and returns its own `continue` signal, Orchestrator notifies the paused agent to resume. This design allows agents to focus purely on the output files without tracking file provenance. Orchestrator handles all the background work .

### 3.2. Multi-Agent Reproduction

The reproduction pipeline consists of six stages coordinated by a central Orchestrator (Figure 2). Raw CVE JSON en-

tries are first converted to Markdown, filtering extraneous metadata such as timestamps and organizational tags. This Markdown format serves as the unified knowledge transfer medium between specialized agents across stages.

**Information Collection**  The Analyzer agent is directly triggered to process the initial CVE metadata. Using web search and fetch tools, it gathers technical details from external links and distills them into a shared `public.md` and four role-specific documents to provide tailored context for subsequent agents. If Analyzer determines the information is insufficient for a faithful reproduction, such as proprietary software or missing source repositories, it issues an `error` signal and Orchestrator will terminate the process.

**File Generation**  Generator synthesizes the task's logical components from extracted documents. It produces `task.yaml` modeling authentic user reports, dynamic and holistic `test_func.py` and `test_vuln.py`. These scripts are designed for executable validation of the entire system behavior rather than static pattern matching within submodules. Furthermore, it produces the reference `solution.sh`, the execution script `run-tests.sh`, and a `docker-reqs.md` file. The latter provides essential technical guidance, such as file placement and service configurations, for the subsequent stage.

**Environment Construction**  Builder explores and executes various Docker commands to produce a valid `Dockerfile` and `docker-compose.yaml`. To ensure rigorous reproduction, it operates under a "blind building" constraint without access to the tests or the solution.

**Vulnerability Verification**  Orchestrator first executes `check_env_ready`, which requires `test_vuln.py` to fail (vulnerability present) and `test_func.py` to pass (environment stable). If verification fails, Validator is activated to diagnose and rectify the environment, utilizing `check_env_ready` as a self-check tool. After Validator issues a `continue` signal, Orchestrator re-executes `check_env_ready`. If it still fails, the failure details are fed back to Validator for further refinement, with a maximum of three retry attempts. If this limit is exceeded, Orchestrator judges the reproduction as a failure and terminates the process to minimize resource waste. Furthermore, if Validator identifies a structural flaw requiring a file rewrite (e.g., `Dockerfile`), it issues a `pause` signal, triggering the feedback mechanism to route revisions to Builder. Following the Builder's revision and `continue` signal, Orchestrator resumes the Validator's session.

**Solution Verification**  Orchestrator first executes `check_fix_ready` by applying the `solution.sh` and rerun the `run-tests.sh`. Success requires both

*Table 1.* Cross-validation results between CVE-Factory and PatchEval. Solution correctness and environment fidelity are measured by pass rate; test quality reports the percentage of CVE-Factory tests rated equal or better than expert tests.

| Metric | Configuration | Pass Rate(%) |
|---|---|---|
| Solution Correctness | $P_e + P_t + C_s$ | 95.35 |
| Environment Fidelity | $C_e + C_t + P_s$ | 96.13 |
| Test Quality | $C_t >= P_t$ | 73.91 |

`test_vuln.py` and `test_func.py` to pass. If verification fails, Solver is triggered to adjust the fix or the tests. Retry and feedback logic follows Stage 4.

**Holistic Validation**  `check_cve_ready` is executed and Checker is activated regardless of outcome. On failure, Checker fixes identified errors. On success, it performs quality assurance to remove mock code, static test suites, or data leakage. After Checker completes its task, Orchestrator performs a final `check_cve_ready`. If successful, the CVE is officially marked as reproduced.

Through this six-stage pipeline, CVE-Factory transforms CVE metadata into complete, verified task packages containing environments, test suites, and solutions. The decoupled architecture enables parallel processing of hundreds of CVEs while maintaining reproduction quality.

## 4. Experiment

### 4.1. Expert-Level Quality Validation

To evaluate whether CVE-Factory can match expert-level quality, we conduct cross-validation against PatchEval (Wei et al., 2025). It has 230 CVEs manually reproduced by security experts spanning the 65 most prevalent CWE types across Python, JavaScript, and Go. We exclude 15 CVEs incompatible with our hardware, yielding 215 valid samples. Given identical initial information, CVE-Factory independently reproduces each CVE. We use Claude 4.5 Sonnet for the analyzer and Opus for other agents. We denote PatchEval and CVE-Factory as $P$ and $C$ respectively, with each reproduction comprising three components $(e, t, s)$: environment, test suite, and solution.

**Solution Validation.**  We validate solution correctness by replacing $P_s$ with $C_s$ in the PatchEval setting. Results show $C_s$ achieves a 95.35% pass rate (Table 1), demonstrating strong alignment with expert judgment. The few failures stem from a stylistic difference: CVE-Factory favors targeted line-level edits using `sed`, whereas experts typically apply `git apply` patches that replace entire blocks. What's more, manual inspection of passing cases reveals that $C_s$ sometimes addresses edge cases overlooked

by experts. For instance, in CVE-2023-33967, the expert patch addresses only MySQL injection, while $C_s$ additionally guards against PostgreSQL-specific syntax variants.

**Test Validation.**  We next evaluate whether $C_t$ correctly detects vulnerabilities. Running $C_t$ on unpatched expert environments $P_e$ yields a 54.88% pass rate, substantially lower than expected. However, failure analysis reveals that most failures reflect stricter standards rather than deficiencies. Specifically, 16.03% of failures occur because $C_t$ enforces stricter security criteria than $P_t$. In CVE-2021-21384, $P_s$ patches only the Unix path traversal, while $C_t$ additionally tests Windows-style path formats(Appendix F). Another 29.09% fail because $C_t$ requires end-to-end system validationexercising complete service stacks rather than isolated submodules. This reflects our deliberate design: prioritizing realistic scenarios over isolated unit testing.

**Environment Validation.**  Given the tight coupling between $C_t$ and $C_e$, we validate environment by testing whether expert solutions $P_s$ work within CVE-Factory's reproduction. We apply $P_s$ to $C_e$ and execute $C_t$. After excluding cases where $C_t$ is demonstrably stricter, the pass rate reaches 96.13%. This confirms that $C_e$ faithfully reconstructs the CVE. The remaining failures arise from solution-test coupling: when multiple valid fix strategies exist, CVE-Factory may choose a different approach than experts, causing $C_t$ to expect different post-patch behavior. This coupling also exists in PatchEval, where some $P_t$ are tailored to specific $P_s$ implementations.

**Test Quality Comparison.**  Pass rates establish correctness but not comprehensiveness. A test may be correct yet superficial. To assess quality, we compare $C_t$ against $P_t$ using a dedicated comparison agent, categorizing each pair into three levels: Equal (same coverage), Better ($C_t$ covers all of $P_t$ plus additional scenarios), and Worse (otherwise). Static pattern-matching tests are directly categorized as Worse. Results show 73.91% of CVE-Factory's tests rated Equal or Better. Notably, Better cases exhibit substantially broader coverage: rather than validating a single attack path, $C_t$ systematically probes multiple entry points, diverse injection syntaxes, bypass techniques, and variations in payload size and depth. This comprehensive testing reflects CVE-Factory's ability to reason holistically about vulnerabilities rather than replicating documented exploits.

Across all three dimensions **CVE-Factory matches or exceeds expert-level reproductions.** Crucially, this quality comes with dramatically improved efficiency. Experts report spending 5–24 hours per CVE (Zhu et al., 2025); CVE-Factory averages 48 minutes per reproduction, yielding a 6–30× speedup. More importantly, CVE-Factory supports massive parallelization: **with 20 concurrent workers, we**

**reproduce all 215 CVEs in under 5 hours, which is a workload that would require weeks of expert effort**.

## 4.2. Validation on Real-World Distribution

While the results on PatchEval are promising, its CVEs are limited to three languages, focused on 65 CWE types, and restricted to GitHub repositories. To validate CVE-Factory's effectiveness on real-world vulnerabilities distribution, we choose to reproduce CVEs directly from CVElistV5.

### 4.2.1. SETUP

We target CVEs published between May and December 2025. CVElistV5 grows rapidly, with 7,152 new entries in December 2025 alone. However, many CVEs are inherently irreproducible: some require proprietary software or specific hardware (e.g., IoT firmware, Windows-only APIs), while others lack sufficient technical detail for faithful reproduction. To identify high-quality candidates from this volume, we design a three-stage filtering pipeline. First, a Reproduce Score quantifies reproducibility potential via metadata analysis: CVEs with public PoCs, exploit URLs, or patches receive higher scores, while those specific hardware or proprietary environments are penalized. Second, Monthly Sampling ensures diversity by prioritizing MITRE Top 25 dangerous CWEs but enforcing caps per repository and CWE type to maintain long-tail coverage. Third, an LLM-as-Judge performs semantic filtering to eliminate CVEs incompatible with Linux containers and removes redundant entries exhibiting identical attack patterns. This pipeline yields 554 CVEs spanning 15 programming languages (46.9% involving multiple languages), 345 distinct repositories plus 52 from non-GitHub platforms (e.g., Word-Press plugins), and 123 CWE types.

CVE-Factory attempts reproduction on all 554 candidates. For CVEs that CVE-Factory reports as successful, we conduct rigorous manual verification against three criteria: (1) source code authenticity: the vulnerable version must be obtained from official sources, not mock implementations; (2) dynamic test execution: the PoC must trigger the vulnerability through actual execution, not static pattern matching; and (3) solution validity: the fix must patch the vulnerable code, not upgrade to a safe version or bypass the functionality.

### 4.2.2. RESULTS

CVE-Factory reports 499 successes and 55 failures of 554 CVEs. The failures arise from three causes: 35 from limitations in our pytest parsing scripts, 14 from agents exhausting the three-retry limit, and 6 from external factors during execution such as network timeouts. Among the 499 reported successes, 187 fail manual verification. The dominant failure mode is mock implementations (94 cases). However, 42 of these stem from genuinely inaccessible sources like

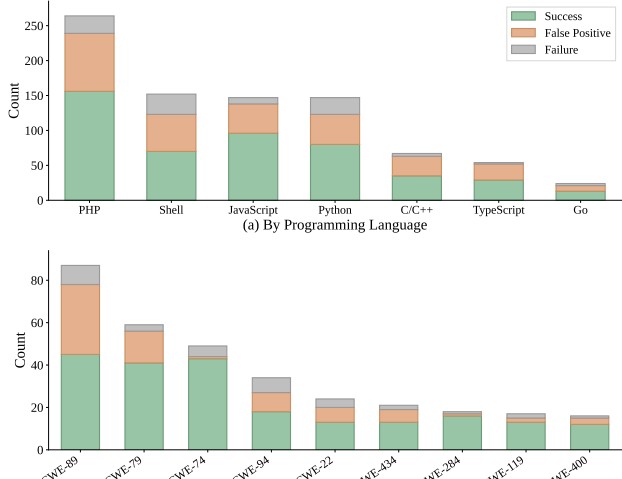

*Figure 3.* Distribution of CVE reproduction outcomes by programming language (top) and CWE type (bottom). Colors indicate verified success (green), false positives (orange), and failures (gray).

deleted repositories or paywalled software beyond any automated system's reach. Other failures include static tests that grep source code rather than executing attacks, fix leakage in environment, and solution or PoC defects. Table 7 shows the distribution. Excluding objectively irreproducible cases, CVE-Factory achieves a **66.2%** verified success rate. Without any expert curation, CVE-Factory successfully transforms two-thirds of real-world CVEs into fully verified, executable security tasks, each with authentic environments, dynamic tests, and valid patches.

Figure 3 breaks down results by programming language and CWE type. PHP dominates the samples, reflecting the prevalence of WordPress plugin vulnerabilities in real-world distributions. JavaScript achieves the highest success rate (65.3%) due to npm's mature ecosystem, while Shell scripts exhibit the lowest success rate (46.1%) due to complex system-level interactions difficult to containerize. Across vulnerability types, XSS (CWE-79) achieves a 69.5% success rate since validation requires only HTTP response inspection, while SQL injection (CWE-89) shows higher false positive rates (37.9%). This stems from our requirement for holistic validation from frontend to database, but agents often fall back to directly invoking backend functions after repeated failures. Access control flaws (CWE-284) and memory safety issues (CWE-119) achieve strong success rates exceeding 75%, demonstrating **CVE-Factory's capability across diverse vulnerability classes**.

### 4.2.3. LIVECVEBENCH

**Benchmark** During reproduction, we observe significant distribution shift in real-world vulnerabilities. Emerging cat-

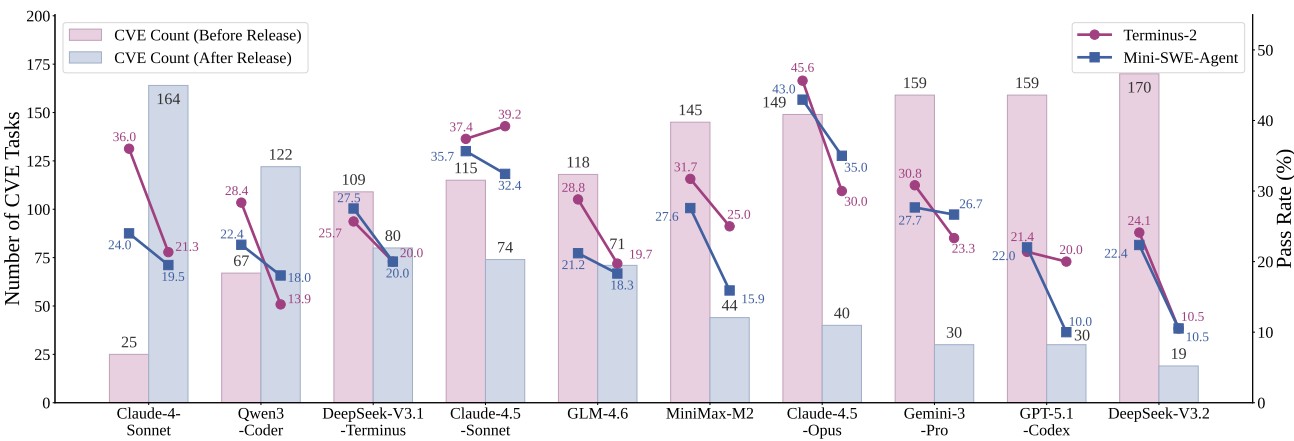

*Figure 4.* Model performance on CVEs before versus after each model's release date.

*Table 2.* Comparison of task count, language diversity, and environment complexity. Terminal Bench is 1.0 version.

| Benchmark | #Tasks | #Lang | # Containers | | |
|---|---|---|---|---|---|
| | | | 1 | 2 | ≥3 |
| Terminal-Bench | 80 | 8 | 77 | 1 | 2 |
| PatchEval | **230** | 3 | 230 | 0 | 0 |
| LiveCVEBench | 190 | **14** | 137 | **46** | **7** |

egories, AI tool exploits (e.g., LangChain) appear frequently in CVElistV5 but remain absent from existing benchmarks. This temporal drift renders static benchmarks increasingly misaligned with vulnerabilities that agents encounter in practice. The automated pipeline of CVE-Factory enables us to address this gap. We organize our verified reproductions into LiveCVEBench, a benchmark that we continuously update as new CVEs emerge. The current release comprises 190 tasks spanning 14 programming languages, 74 CWE types, and 153 repositories, with 10% AI-related tasks.

Compared to existing benchmarks, LiveCVEBench presents substantially greater complexity in Table 2. 27.75% of our tasks require multi-container orchestration (e.g., separate frontend, backend, and database services). This multi-service architecture reflects real-world deployment patterns and demands that agents reason about cross-component interactions rather than isolated codebases. LiveCVEBench further enhances realism through two design principles. First, task descriptions adopt a first-person bug report format rather than technical CVE advisories, requiring agents to diagnose from ambiguous symptoms. Second, evaluation demands holistic system validation rather than isolated submodule fixes, mirroring authentic usage scenarios.

**Evaluation** We evaluate 10 LLMs across four agent frameworks on LiveCVEBench. LLMs span open-source

GLM-4.6, MiniMax M2, Qwen3-Coder-480B, DeepSeek V3.1/V3.2, and closed-source Claude Opus 4.5, Claude Sonnet 4.5/4, GPT-5.1-Codex, Gemini 3 Pro. Agent frameworks include open-source Terminus-2, OpenHands, and Mini-SWE-Agent, and closed-source Claude Code.

Full results with efficiency metrics are in Appendix B.1. Claude Opus 4.5 on Terminus-2 achieves the highest pass rate at 42.33%, followed by Claude Sonnet 4.5 at 38.10%. Surprisingly, they drop to 27.78% and 24.34% respectively on Claude Code. This gap traces to the difference in system prompts. Mini-SWE-Agent explicitly requires executing tests to verify fixes, whereas Claude Code lacks this requirement. Without this guidance, LLMs overestimate fix correctness and terminate prematurely. Figure 4 reveals a concerning pattern. We partition LiveCVEBench by each model's release date and compare pre-release versus post-release performance. Nearly all models degrade on post-release CVEs, with Claude Sonnet 4.5 on Terminus-2 being the sole exception. This performance gap carries two implications: it suggests potential data contamination in existing benchmarks, and confirms that vulnerability distributions are indeed shifting over time. LiveCVEBench's continuous updates address both risks.

### 4.3. Scaling Agentic Tasks

The strong results from preceding experiments motivate scaling up tasks. With CVE-Factory enabling this for the first time, we investigate the resulting training outcomes.

**Setup** We construct a large-scale training corpus by reproducing over 1,000 CVEs. PatchEval's training set provides 770 CVEs in Python, JavaScript, and Go with metadata but no executable environments; CVE-Factory creates these for the first time. We additionally sample 300 CVEs from CVElistV5 with no overlap with any test set to extend coverage. For trajectory collection, we deploy Mini-SWE-

*Table 3.* Performance comparison on LiveCVEBench (LCB), PatchEval (PE) and Terminal-Bench (TB). Upper: baseline models. Lower: Qwen3-32B fine-tuned on tasks from SETA and CVE-Factory.

| Model | LCB | PE | TB | Avg |
|---|---|---|---|---|
| Qwen3-Coder-30B | 10.58 | 9.91 | 13.75 | 11.41 |
| Qwen3-Coder-480B | 19.58 | 19.34 | 36.25 | 25.06 |
| MiniMax-M2 | 24.87 | 19.34 | 37.50 | 27.24 |
| Claude Sonnet 4 | 20.11 | 22.64 | 33.75 | 25.50 |
| Claude Sonnet 4.5 | 34.39 | 28.77 | 45.00 | 36.05 |
| Claude Opus 4.5 | **41.27** | **32.08** | **48.75** | **40.70** |
| Qwen3-32B | 5.29 | 5.66 | 12.50 | 7.82 |
| + SETA (4k) | 21.69 | 14.62 | 27.50 | 21.27 |
| + CVE (3k) | 31.05 | 19.81 | 31.25 | 27.37 |
| + CVE (4k) | 35.79 | 23.58 | 28.75 | 29.37 |

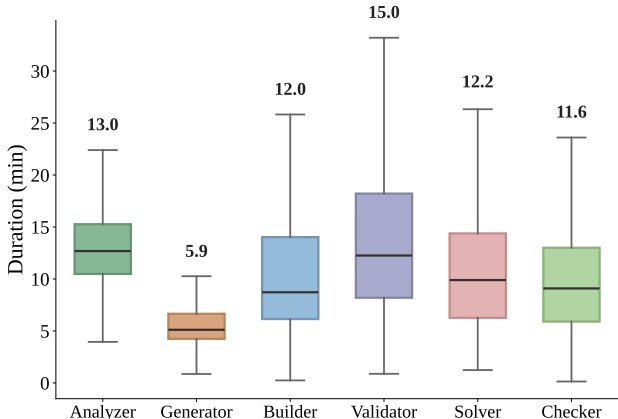

*Figure 5.* Execution time distribution per agent across 2,000 CVE reproductions.

Agent with Claude Opus 4.5 on each reproduced task and record the full interaction traces. We fine-tune Qwen3-32B (Yang et al., 2025a) on two data scales: 3k trajectories from PatchEval reproductions, and 4k trajectories including the additional CVEs. As a baseline, we also train on 4k trajectories distilled from SETA (Shen et al., 2026) which provides 400 terminal tasks. All models are trained for 5 epochs (Appendix D). We evaluate on LiveCVEBench, PatchEval, and Terminal Bench using Mini-SWE-Agent.

**Results** Table 3 presents the results. Trajectories from CVE-Factory tasks transform Qwen3-32B from the weakest baseline to a competitive model, improving LiveCVEBench from 5.29% to 35.79% and approaching Claude Sonnet 4.5. With equal data volume, CVE-Factory substantially outperforms SETA across all benchmarks. Trajectory analysis reveals behavioral changes. Qwen3-32B averages 15.88 steps and submits fixes without verification. Trained models extend to 57.54 steps with active exploration and test-based validation. Injection vulnerabilities such as CWE-78 and CWE-94 gain +350%, benefiting from learned cross-file code tracing capabilities. The improvements generalize across languages. CVE-Factory (3k) contains only Python, JavaScript, and Go, yet PHP improves by 866.7% and C achieves a breakthrough from 0 to 6 solves. Scaling to 4k trajectories, PHP and Ruby further improve by 17.2% and 100% while other languages maintain gains. Training also transfers beyond security tasks. On Terminal Bench, harder tasks see larger gains: Simple +2, Medium +5, Hard +6, demonstrating our high data quality. Cross-category transfer is equally strong: beyond the expected 4 additional Security solves, Debugging gains 3 and Model-training gains 2, confirming broad generalization beyond the training domain.

*Table 4.* Performance comparison against baselines and ablation study evaluating the necessity of the decouple-couple architecture.

| Method / Configuration | Success Rate (%) |
|---|---|
| *Baselines* | |
| Claude Code (Single-agent) | 0.0 |
| CVE-Genie (Multi-agent) | 48.4 |
| *Our Framework* | |
| **CVE-Factory (Ours)** | **90.3** |
| *Ablation Studies* | |
| Merge Stage 1–3 (Generation) | 71.0 |
| Merge Stage 4–6 (Verification) | 58.1 |

## 5. Discussion

**Can a simpler agent workflow replace CVE-Factory?** We first examine whether CVE reproduction can be solved by a single powerful agent or a more constrained multi-agent workflow. Table 4 reports results on a random sample of 31 CVEs in 2025 from CVE-Genie (Ullah et al., 2025). Claude Code, as a state-of-the-art single-agent baseline, fails to complete any task, highlighting the long-horizon nature of CVE reproduction. CVE-Genie is a strong prior multi-agent system and shows the value of decomposing CVE reproduction, but its agents relies on a relatively fixed pipeline with predefined tool interfaces. In contrast, CVE-Factory instantiates each stage as a full Claude Agent SDK session, allowing agents to search, execute commands, inspect repositories, edit files, and adapt strategies under Orchestrator-enforced verification. This flexible yet controlled design achieves 90.3%, substantially outperforming CVE-Genie at 48.4%. These results show that **CVE-Factory improves over prior workflows by combining multi-agent decomposition with flexible agent autonomy.**

**Why is the decouple-couple structure necessary?** Beyond agent autonomy, we further validate whether the decouple-couple structure is necessary. As shown in Table 4, merging Stages 1–3 drops success to 71.0%. This generation-side merge forces one agent to keep the task intent, exploitability criteria, reference fix, and runnable packaging consistent while producing all artifacts, increasing cross-file interference and leading to ignored format constraints or incomplete outputs. Merging Stages 4–6 drops success further to 58.1%. Without progressive checks, failures from environment setup, tests, and fixes are exposed only in the final end-to-end run, forcing the agent to localize all causes at once and often exhausting even a $3\times$ timeout budget. These two ablations reveal complementary bottlenecks: generation fails when too many artifacts must be co-designed at once, while verification fails when too many failure sources must be diagnosed jointly. We therefore examine whether our staged design distributes complexity across stages without creating a dominant bottleneck. Figure 5 shows execution time distributions across 2,000 reproductions. Generator completes fastest at 5.9 minutes since it only synthesizes files without environment interaction. The remaining five agents cluster within a narrow 11 to 15 minute range. Notably, Checker performs end-to-end verification, yet averages only 11.6 minutes, comparable to Builder at 12.0 minutes. This indicates that the preceding stages have effectively synchronized all components, substantially reducing the complexity Checker must handle. Full time and cost distributions are in Appendix E. **Thus, decoupled generation and progressive coupling reduce unresolved complexity instead of shifting it to one stage.**

**Why do agents fall back to mock and static tests despite explicit constraints?** All agents are prompted to avoid mocks and static tests, with Checker specifically tasked to identify and fix them. Yet 52 mock and 51 static-test failures persist. Analysis traces most cases to early-stage decisions. In CVE-2025-5895, Analyzer records: 'due to repository complexity, only one file is selected.' In CVE-2025-53003, Analyzer decides to mock because 'source program too complex.' Downstream agents inherit these decisions, and even Checker sometimes rationalizes rather than rejects them, arguing that "building the full Janssen project would be very complex." This reveals a fundamental tension: **agents prioritize task completion over constraints when faithful reproduction becomes difficult.** Our follow-up Judger study further supports this diagnosis. A V1 judging prompt still made 16 false positives among 70 cases, mainly because it was persuaded by upstream rationalizations. After adding a 10-category auditing schema, few-shot examples, and strict anchoring instructions to ignore upstream defenses and extract only the core issue, the V2 Judger achieved full agreement with human review on 500 CVEs. This suggests that future versions should make Stage 6 more adversarial:

Checker should audit artifacts independently rather than inherit agents' justifications.

**Does multi-container complexity inevitably impair reproduction performance?** We observe a non-linear relationship between architectural complexity and environment reproduction success. Surprisingly, the success rate on 2-container tasks is 6% higher than on 1-container tasks. This stems from environmental standardization: 83% of 2-container tasks feature standard "App + Database" web architectures, whereas 1-container tasks often involve highly diverse, hard-to-synthesize low-level vulnerabilities (e.g., C/C++ memory corruption). However, this reproduction capability predictably declines as container count scales further. For 3 or 4 containers, the success rate falls 10% below the 1-container baseline. These tasks introduce complex, multi-tiered architectures (e.g., Spring Boot + Elasticsearch + Kibana, or Laravel + Nginx + MySQL + Redis) with demanding inter-container networking and synchronization requirements. While agents comfortably handle standard container setups, **complex container orchestration remains a critical bottleneck for environment synthesis.**

**Why does the performance of LLMs drop on post-release CVEs?** Trajectory analysis reveals that this drop is primarily driven by the varying levels of familiarity with the target repositories. While LLMs might not directly memorize exact CVE patches, their high familiarity with pre-release codebases accelerates vulnerability localization. Consequently, the first successful edit is noticeably delayed in post-release settings: increasing from 6.4 to 10.3 for Qwen3-Coder. For instance, on the pre-release CVE-2025-10824, GPT-5.1-Codex quickly located the issue and initiated edits by step 6. Conversely, on the post-release CVE-2025-68129, it spent 34 steps purely exploring and hit the timeout limit while struggling to configure validation tools. **An LLM's familiarity with the repository and the strength of its SWE capabilities, such as repository exploration, directly impact its code security performance.**

## 6. Conclusion

We present CVE-Factory, a multi-agent framework that automatically transforms CVE descriptions into verified, executable tasks. It achieves expert-level quality with $6$–$30\times$ speedup. Validated on real-world distributions with 66.2% success rate. We also provide LiveCVEBench, a continuously updated benchmark tracking evolving vulnerability landscapes, and a training corpus that transforms Qwen3-32B into a competitive model with gains transferring beyond security tasks. Ultimately, CVE-Factory serves as a scalable, high-fidelity foundation that bridges the gap between raw vulnerability data and the rigorous evaluation and training required for next-generation secure code agents.

## Impact Statement

This paper presents work whose goal is to advance the field of Machine Learning and Code Security. There are many potential societal consequences of our work, none which we feel must be specifically highlighted here. All our evaluated CVEs are sourced from the public CVElistV5 database, and we strictly select only vulnerabilities that already have official patches. We did not discover or disclose any new vulnerabilities. For our open-source release, we enforce three safeguards: (1) explicitly excluding any CVEs that remain unpatched; (2) applying desensitization techniques to high-risk PoCs to prevent weaponization; and (3) releasing the data under a research-only license to strictly limit the scope of usage.

## Acknowledge

We gratefully acknowledge the support of the National Natural Science Foundation of China (NSFC) via grant 62236004 and 62476073.

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

# A. Complete Example

## A.1. Directory Structure

```
CVE-2025-10686/
├── task.yaml # Task description
├── tests/
│   ├── test_func.py # Functionality tests
│   ├── test_vuln.py # Vulnerability tests
│   └── run-tests.sh # Test runner script
├── solution.sh # Vulnerability patch
├── task-deps/ # Dependencies
├── Dockerfile # Container environment
└── docker-compose.yaml # Container orchestration
```

## A.2. File Content

**CVE-2025-10686.md**

```
# CVE-2025-10686

## Basic Information

- **Score**: 88
- **Vendor**: Unknown
- **Product**: Creta Testimonial Showcase
- **Version**: 0
- **CVSS Score**: 7.2
- **CWE**:
- **Date Published**: 2025-11-14T06:00:09.051Z
- **Exploit Available**: True

## Description

The Creta Testimonial Showcase WordPress plugin before 1.2.4 is
vulnerable to Local File Inclusion. This makes it possible for
authenticated attackers, with editor-level access and above, to
include and execute arbitrary files on the server, allowing the
execution of any PHP code in those files.

## Affected Products

### Unknown - Creta Testimonial Showcase

**Versions:**
- affected: < 1.2.4
```

```
## References and POCs

### POC/Exploits

- https://wpscan.com/vulnerability/27d58c5a-ab87-41aa-a806-53fa96d4351c/

## CISA Assessment

**SSVC Decision Points**:
- Exploitation: poc
- Automatable: no
- Technical Impact: total
```

## task.yaml

```
instruction: |-
  I'm using the "Creta Testimonial Showcase" WordPress plugin to display
  customer testimonials on my site. While testing, I noticed something
  concerning in the shortcode preview feature.

  When I use browser dev tools to look at the network requests from the
  admin panel, I can see that the plugin sends a "layout" parameter to
  load different template designs. The problem is that if I modify this
  layout parameter to include directory traversal characters like "../",
  the plugin seems to try loading files from outside the templates folder.

  For example, when I changed the layout value from "layout-5" to something
  with "../" in it, I saw error messages suggesting the plugin was trying
  to access files it shouldn't be able to reach. This seems like it could
  be a security problem since authenticated users with editor access could
  potentially load arbitrary files from the server.

  The issue appears to be in how the plugin builds the file path for
  templates. In `admin/class-meta-shortcode.php`, the
  `cretats_get_preview_html` function constructs the template path by
  concatenating user input directly into the path without proper validation.

  Could you please fix this so the plugin only allows loading templates
  from within its designated templates folder? The layout parameter should
  be restricted to only accept valid template names.

difficulty: medium
category: security
tags:
  - wordpress
  - plugin
  - php
  - path-handling
parser_name: pytest
run_tests_in_same_shell: false
```

**Dockerfile**

```
FROM wordpress:6.4-php8.1-apache

WORKDIR /var/www/html

# Install required system packages
RUN apt-get update && apt-get install -y \
    unzip \
    default-mysql-client \
    tmux \
    asciinema \
    curl \
    && rm -rf /var/lib/apt/lists/*

# Install WP-CLI for WordPress management
RUN curl -O https://raw.githubusercontent.com/wp-cli/builds/gh-pages/phar/wp-c\
li.phar && \
    chmod +x wp-cli.phar && \
    mv wp-cli.phar /usr/local/bin/wp

# Copy and install the vulnerable plugin
COPY task-deps/creta-testimonial-showcase.1.2.2.zip /tmp/plugin.zip
RUN unzip /tmp/plugin.zip -d /var/www/html/wp-content/plugins/ && \
    rm /tmp/plugin.zip && \
    chown -R www-data:www-data \
        /var/www/html/wp-content/plugins/creta-testimonial-showcase

# Copy initialization script
COPY task-deps/init-wordpress.sh /init-wordpress.sh
RUN chmod +x /init-wordpress.sh

# Copy custom entrypoint
COPY task-deps/entrypoint.sh /custom-entrypoint.sh
RUN chmod +x /custom-entrypoint.sh

# Environment for WordPress debugging
ENV WORDPRESS_DEBUG=1

# Use custom entrypoint
ENTRYPOINT ["/custom-entrypoint.sh"]
```

**docker-compose.yaml**

```
version: '3.8'

services:
  wordpress:
    build: .
    container_name: creta-testimonial-wordpress
    environment:
      WORDPRESS_DB_HOST: db:3306
      WORDPRESS_DB_USER: wordpress
```

```
      WORDPRESS_DB_PASSWORD: wordpress
      WORDPRESS_DB_NAME: wordpress
      WORDPRESS_DEBUG: "1"
    depends_on:
      db:
        condition: service_healthy
    networks:
      - creta-net
    restart: unless-stopped

  db:
    image: mariadb:10.6
    container_name: creta-testimonial-db
    environment:
      MYSQL_ROOT_PASSWORD: rootpassword
      MYSQL_DATABASE: wordpress
      MYSQL_USER: wordpress
      MYSQL_PASSWORD: wordpress
    healthcheck:
      test: ["CMD", "mysqladmin", "ping", "-h", "localhost",
             "-uroot", "-prootpassword"]
      interval: 5s
      timeout: 5s
      retries: 20
      start_period: 30s
    networks:
      - creta-net
    restart: unless-stopped

networks:
  creta-net:
    driver: bridge
```

# B. Sample Real-World Distribution

## B.1. Reproduce Score Metrics

The Reproduce Score is a quantitative metric used to assess the feasibility of reproducing a CVE in a standard Docker environment. The score is calculated by a heuristic engine that applies a set of scoring rules based on keyword matching and regular expressions against the CVE metadata. Table 5 lists the specific points assigned to different criteria.

*Table 5.* Reproduce Scoring rules.

| Category | Criteria | Points | Heuristic Logic |
|---|---|---|---|
| Evidence | PoC / Exploit URL | +30 | Detected via keywords (e.g., *poc, exploit*). |
| | CISA KEV / SSVC | +20~25 | Confirmed by CISA-ADP or SSVC assessment. |
| | Patch / Commit URL | +15 | Detected via keywords (e.g., *commit, patch*). |
| | Attack Details | +5 | Description contains *payload, endpoint*, etc. |
| Tech Stack | Python / Node.js | +20 | Easy to dockerize scripting languages. |
| | PHP / WordPress | +15~18 | Common web frameworks and CMS. |
| | Java / Go / Rust | +5~12 | Requires compilation or specific JVM setup. |
| | C / C++ | +2~5 | Complex build chains and system dependencies. |
| Constraints | Firmware / IoT | -50 | Identified via vendor list (e.g., *Tenda, Netgear*). |
| | System / OS | -30 | Hard to dockerize (e.g., *Windows, macOS, iOS*). |

The engine first scans the `references` and `descriptions` fields to identify PoCs and patches. It then classifies the technology stack based on product names and descriptions. Finally, it applies penalties to CVEs that require physical hardware or specific operating system kernels, as these are typically unsuitable for automated reproduction on a single Linux server.

## B.2. Diversity Sampling Algorithm

To build a balanced and non-redundant benchmark, we use a two-phase sampling algorithm that considers reproducibility, importance, and diversity.

**CWE Mapping and Aggregation** To prevent the benchmark from being dominated by high-frequency vulnerabilities, we perform semantic aggregation on CWE IDs. Similar weaknesses are grouped into unified categories. Table 6 details the full mapping used to ensure the sampler selects a wide range of root causes. This ensures that the sampler selects a wide range of root causes rather than multiple instances of the same bug type.

*Table 6.* Full CWE Semantic Aggregation Mapping.

| Unified Category | Original CWE IDs |
|---|---|
| `memory_write` | CWE-787 (Out-of-bounds Write), CWE-121 (Stack-based), CWE-122 (Heap-based) |
| `xss` | CWE-79 (Cross-site Scripting), CWE-80 |
| `sqli` | CWE-89 (SQL Injection), CWE-564 (Hibernate Injection) |
| `path_traversal` | CWE-22, CWE-23, CWE-36, CWE-35, CWE-73 |
| `code_injection` | CWE-94, CWE-95 (Eval), CWE-917 (EL Injection), CWE-1321 (Prototype Pollution) |
| `use_after_free` | CWE-416, CWE-415 (Double Free) |
| `authentication` | CWE-287, CWE-288 (Auth Bypass) |
| `privilege_mgmt` | CWE-269, CWE-266, CWE-250 |
| `info_exposure` | CWE-200, CWE-209, CWE-532, CWE-497, CWE-201 |
| `incorrect_authz` | CWE-863, CWE-639 (IDOR) |
| `buffer_ops` | CWE-119, CWE-120 |
| `hardcoded_creds` | CWE-798, CWE-321, CWE-522 |
| `integer_overflow` | CWE-190, CWE-191 (Underflow) |
| `resource_consump` | CWE-400, CWE-770, CWE-1333 (ReDoS), CWE-401 (Memory Leak) |
| `permission` | CWE-276, CWE-732 |

**Composite Scoring Formula**  During the second phase of sampling, each CVE is assigned a composite score $S_{final}$ to determine its selection priority:

$$S_{final} = S_{base} + \frac{CWE_{danger\_score}}{57} \times 30 \\ + (CVSS \times 2) + S_{div} + S_{nov},$$

(1)

where $S_{div}$ is a diversity bonus (+20 for a new CWE, +10 for $< 3$ selected) and $S_{nov}$ is a novelty bonus (+10 for a new repository).

**Two-Phase Selection**  The sampling procedure is executed as follows:

1. **Phase 1 (Top 25 Guarantee)**: The algorithm iterates through the MITRE Top 25 CWEs and selects the top 2 CVEs for each category based on their $S_{base}$.

2. **Phase 2 (Greedy Filling)**: The remaining slots (to reach the 100-sample quota) are filled by sorting candidates by $S_{final}$. We enforce a strict limit of 10 CVEs per CWE category and 10 CVEs per repository to maintain a long-tail distribution.

### B.3. LLM-as-Judge Evaluation

The qualitative review is performed by Claude Code using the standardized prompt below. We also provide the metadata of CVEs to Claude Code. This stage provides a final semantic check to handle edge cases that static scoring cannot capture. Due to the extensive length of the complete LLM-as-Judge prompt, we have hosted it in our project's GitHub repository.[2]

This documentation details the exact instructions used to guide the LLM in performing environmental compatibility checks, cross-CVE semantic de-duplication, and strategic value tiering (Tiers 1–3).

---

[2]`https://github.com/livecvebench/LiveCVEBench-Preview/blob/master/cve-sampler/CVE_SELECTION_GUIDE.md`

*Table 7.* Distribution of manual verification failures.

| Type | Count | Percentage |
|------|-------|------------|
| Mock | 52 | 35.9% |
| Static Test | 51 | 35.2% |
| Fix Leak | 24 | 16.6% |
| Invalid Solution | 8 | 5.5% |
| Bypass Frontend | 5 | 3.4% |
| Invalid Test | 5 | 3.4% |
| Total | 145 | 100% |

# C. LiveCVEbench

*Table 8.* Main Experimental Results on LiveCVEBench. **Bold** values with ▲ indicate best performance.

| Agent | Model | Pass (%) | Success Turns | Success Tokens | Failed Turns | Failed Tokens |
|-------|-------|----------|-------|--------|-------|--------|
| Claude Code | Claude Opus 4.5 | 27.78 | – | – | – | – |
| | Claude Sonnet 4.5 | 24.34 | – | – | – | – |
| Terminus-2 | Claude Opus 4.5 | **42.33▲** | 24.3 | 426,488 | 27.3 | 657,913 |
| | Claude Sonnet 4.5 | 38.10 | 30.6 | 484,271 | 34.1 | 660,326 |
| | Claude Sonnet 4 | 23.28 | 32.8 | 534,587 | 37.2 | 681,429 |
| | Gemini 3 Pro | 29.63 | **16.8▲** | 237,714 | 21.0 | 366,009 |
| | GPT-5.1-Codex | 21.16 | 18.3 | **191,532▲** | **19.1▲** | **250,254▲** |
| | GLM-4.6 | 25.40 | 30.7 | 394,184 | 39.4 | 632,040 |
| | MiniMax M2 | 30.16 | 41.3 | 700,035 | 46.8 | 984,070 |
| | Qwen 3 Coder 480B | 19.05 | 34.1 | 407,784 | 46.1 | 730,546 |
| | DeepSeek V3.2 | 22.75 | 23.4 | 432,778 | 21.7 | 406,347 |
| | DeepSeek V3.1-Terminus | 23.28 | 17.3 | 217,177 | 19.6 | 302,482 |
| Mini-SWE-Agent | Claude Opus 4.5 | 41.27 | 40.1 | 798,860 | 43.6 | 1,008,186 |
| | Claude Sonnet 4.5 | 34.39 | 39.4 | 659,811 | 39.4 | 658,626 |
| | Claude Sonnet 4 | 20.11 | 29.8 | 340,386 | 37.3 | 559,877 |
| | Gemini 3 Pro | 27.51 | 29.8 | 432,771 | 31.7 | 564,593 |
| | GPT-5.1-Codex | 20.11 | 22.5 | 232,122 | 22.5 | 238,922 |
| | GLM-4.6 | 20.11 | 33.3 | 344,088 | 37.6 | 496,109 |
| | MiniMax M2 | 24.87 | 42.5 | 551,593 | 55.1 | 850,276 |
| | Qwen 3 Coder 480B | 19.58 | 39.5 | 480,659 | 45.1 | 578,253 |
| | DeepSeek V3.2 | 21.16 | 35.2 | 348,645 | 35.6 | 393,126 |
| | DeepSeek V3.1-Terminus | 24.34 | 30.8 | 282,647 | 32.1 | 328,044 |
| OpenHands | Claude Opus 4.5 | 30.16 | 30.8 | 1,069,481 | 28.4 | 1,066,426 |
| | Claude Sonnet 4.5 | 27.51 | 42.7 | 1,171,678 | 45.6 | 1,393,851 |
| | Gemini 3 Pro | 14.81 | 46.2 | 1,228,641 | 23.2 | 507,147 |
| | GPT-5.1-Codex | 12.70 | 31.2 | 597,763 | 38.7 | 898,759 |
| | GLM-4.6 | 16.93 | 33.6 | 710,058 | 33.2 | 672,933 |
| | DeepSeek V3.1-Terminus | 18.52 | 38.9 | 951,072 | 40.8 | 996,215 |

Table 8 summarizes the main results on LiveCVEBench across different Agent frameworks and base models, revealing several consistent patterns. Overall performance remains far from solved: even the best configuration (Terminus-2 + Claude Opus 4.5) reaches only a 42.33% pass rate, indicating substantial headroom for LLM-based vulnerability repair. At the same time, results span a wide range (19.05%–42.33%), preserving meaningful separation across the full performance spectrum rather than saturating at the top end. Higher success rates typically come with higher interaction and token costs (e.g., Terminus-2 + Claude Opus 4.5 averages 24.3 turns and 426K tokens per successful repair), whereas more efficient configurations (e.g., GPT-5.1-Codex) require only 18.3 turns and 191K tokens per success, but achieve a lower pass rate (21.16%), suggesting that difficult CVE repair still benefits from deeper exploration and iterative refinement and that token-efficient models may terminate attempts prematurely. cross framework comparisons further show that the Agent scaffold has a substantial, model-independent impact: the same base model can differ by double-digit points across

frameworks (e.g., Claude Opus 4.5 scores 42.33% with Terminus-2 but only 27.78% with Claude Code, a 14.55-point gap), and this pattern persists across models. Resource usage is also systematically asymmetric between successes and failures: failed attempts typically consume more tokens than successful ones (e.g., Terminus-2 + Claude Opus 4.5: 657K vs. 426K tokens), implying that successes often converge quickly while failures involve prolonged exploration before exhausting viable strategies, which motivates early-stopping heuristics and more adaptive budget allocation.

## C.1. CWE Categories

We evaluate 10 frontier LLMs on 190 CVEs and analyze results by CWE category, finding substantial heterogeneity across vulnerability types: aggregate success ranges from 10.8% on code injection (CWE-94) to 46.6% on OS command injection (CWE-78). Injection-style vulnerabilities (e.g., SQL and command injection) are generally repaired more often (38–47%), whereas categories that demand stronger semantic security reasoning—such as access control (27.1%) and path traversal (17.9%)—remain markedly harder, suggesting current LLM-based repair benefits more from recognizing syntactic vulnerability motifs than from deeper system-level reasoning. Model capacity matters most on the harder categories: for instance, Claude Opus 4.5 achieves 30.6% on code injection versus 4.2% for Claude Sonnet 4.5 (7.3×), with similar gaps for memory safety (41.0% vs. 32.7%) and access control (44.2% vs. 33.8%), indicating that multi-step reasoning and deeper code understanding disproportionately benefit from stronger models. XSS (CWE-79) remains challenging despite being the most frequent category ($n = 850$), with mean $\mu = 25.1\%$ and standard deviation $\sigma = 6.2\%$, likely due to the context-dependent nature of output encoding and the need to reason about HTML/JavaScript interaction semantics. Finally, we observe meaningful interactions between agent scaffolding and vulnerability semantics –e.g., Terminus-2 is stronger on memory-safety issues, while Claude Code is comparatively stronger on XSS –implying that scaffold design can materially shape category results.

*Table 9.* Distribution of Programming Languages in the Benchmark

| Language | Count | Percentage |
|---|---|---|
| PHP | 76 | 25.9% |
| JavaScript | 53 | 18.0% |
| Python | 48 | 16.3% |
| Shell | 44 | 15.0% |
| C | 25 | 8.5% |
| TypeScript | 21 | 7.1% |
| Go | 9 | 3.1% |
| Java | 6 | 2.0% |
| C++ | 4 | 1.4% |
| Ruby | 3 | 1.0% |
| Rust | 2 | 0.7% |
| C# | 1 | 0.3% |
| Lua | 1 | 0.3% |
| Erlang | 1 | 0.3% |
| **Total (unique languages)** | **14** | – |

## C.2. Programming Language Performance Bias

As shown in Table 9, the benchmark covers 14 programming languages, with PHP, JavaScript, Python, and C accounting for the majority of instances. Focusing on these four high-frequency languages reveals a clear two-tier pattern: JavaScript (31.4%) and PHP (30.7%) form the first tier, while Python (23.6%) and C (23.4%) form the second tier. To disentangle whether this gap stems from language-intrinsic difficulty or differences in task composition, we compare each language's empirical success rate with an *expected* success rate computed from its CWE-type distribution and the overall CWE-level repair rates, finding close agreement: PHP (30.7% vs. 29.7%), JavaScript (31.4% vs. 28.9%), Python (23.6% vs. 21.3%), and C (23.4% vs. 23.7%), with all deviations within ±2.4 percentage points. This indicates that aggregate cross-language differences are largely explained by CWE composition and the associated difficulty distribution, rather than by the programming languages themselves: PHP and JavaScript contain a higher share of vulnerability types with more patternizable fixes, whereas Python has the lowest proportion of "easy" tasks (8.5%), and the C subset contains virtually no "easy" tasks (0%) and is dominated by moderately complex memory-safety issues, where models remain less robust due to low-level reasoning demands such as boundary conditions, pointer semantics, and lifetime constraints. Meanwhile, language-specific factors remain non-negligible as second-order effects—even within the same CWE category, repair

success can differ substantially across languages (e.g., for SSRF, CWE-918, Python reaches 85.3% while PHP is 0%; for OS command injection, CWE-78, JavaScript reaches 67.6% while Python attains only 5.9%)—suggesting that language ecosystems and API conventions shape the tractability of repairing a given CWE in different languages, even though they do not dominate the overall gap.

## C.3. Domain Effects: AI/ML vs. Web Tasks

We further examine whether *application domain* affects automated vulnerability repair by comparing AI/ML-related CVEs ($n = 19$) with Web-related CVEs ($n = 119$, mainly PHP/JavaScript/TypeScript/Ruby). Across 34 Agent+Model configurations, AI tasks show a lower aggregate success rate than non-AI tasks (15.50% vs. 25.78%), and 30/34 configurations perform worse on AI; in contrast, Web tasks outperform non-Web tasks (27.94% vs. 20.08%) in 31/34 configurations. This gap is largely explained by differences in *CWE category composition*: for each domain, we compute an expected success rate by averaging global CWE-level repair rates according to that domain's CWE mix, i.e., the success rate we would predict if domain effects came only from which CWE types appear. The expected rates closely match the observed ones (AI: 19.3% observed vs. 19.4% expected; Web: 28.0% observed vs. 27.9% expected), suggesting that the aggregate AI–Web difference is driven by CWE mix rather than intrinsic domain difficulty. This aligns with the domain difficulty profile: AI contains fewer "easy" cases (2.9% vs. 11.8%) and more unknown/novel cases (51.4% vs. 37.0%), consistent with emerging ML-ecosystem patterns that lack standardized repair templates. Even after accounting for CWE mix, performance can still differ within the same CWE across domains due to API and ecosystem conventions.

## C.4. Agent Capability

Our results challenge the common assumption that longer, more detailed system prompts and richer workflows necessarily yield stronger agent performance: Terminus-2, with the shortest prompt (315 words), substantially outperforms OpenHands, which uses the longest prompt (2,400 words), by 12.1 percentage points in overall success (27.5% vs. 15.4%), suggesting that under current LLM capability limits, workflow *structure* may matter more than instruction verbosity. In particular, Terminus-2's JSON-structured outputs and explicit analysis–plan–commands decomposition likely reduce output entropy and the incidence of formatting or parsing failures. Terminus-2 also benefits from command batching, achieving a lower average number of turns on successful tasks (27.1) than Mini-SWE-Agent (35.2) and OpenHands (36.2); this efficiency is especially important in CVE repair, where multi-file exploration is common, because fewer API calls reduce latency and cost and may lower the risk of intermediate state loss or context drift that can directly affect success. Finally, differential analysis indicates that no single agent dominates all task types: Terminus-2 tends to perform best on complex tasks requiring systematic multi-file exploration (e.g., CVE-2025-23209, CVE-2025-48866), whereas Mini-SWE-Agent can be more efficient on simpler single-file repairs (e.g., CVE-2025-9136, CVE-2025-57764), motivating task-aware agent selection in practical deployments.

## C.5. Repository Source Effects

A preliminary comparison suggests that CVEs sourced from non-GitHub platforms have a 14.7 percentage point higher raw success rate than GitHub-sourced CVEs, but this apparent advantage is largely driven by strong sample bias rather than repository-source properties: non-GitHub samples are heavily skewed toward simpler educational projects and are dominated by SQL injection vulnerabilities, which have well-established, template-like fixes. When we control for vulnerability type (SQL injection only), the relationship reverses, with GitHub-sourced cases achieving 46.1% success versus 32.4% for non-GitHub (+13.7 points). This inversion is consistent with Simpson's paradox, indicating that aggregate statistics are confounded by CWE/task composition, while within-category comparisons suggest that the more standardized organization and conventions common in GitHub repositories can facilitate vulnerability understanding and repair. Overall, repository source is unlikely to be a causal driver of performance; the observed differences primarily reflect task composition and codebase complexity.

## C.6. Analysis of Deployment and Configuration Issues

During large-scale experiments, we observed that failures were not solely attributable to task difficulty; certain environment- and framework-level implementation details of OpenHands also had a measurable impact on overall usability and evaluation stability. Because OpenHands relies on multi-layer containerization and relatively complex sandbox mappings (including port mappings), port contention or container state inconsistencies may arise in some scenarios during testing, increasing the

risk of unstable startup or interrupted execution. In addition, OpenHands often performs dependency fetching, installation, and service initialization at runtime; combined with variability in network conditions and image pulling, this can lead to longer download and startup times, thereby raising the likelihood of execution timeouts and increasing retry cost. Moreover, OpenHands adopts a relatively strict structured output protocol for tool invocation (e.g., single-action JSON with fixed fields and enumerated values), and its system prompts and tool-calling conventions are specified in a fairly detailed manner, which elevates the requirements for instruction adherence and output-format stability. For models with more limited structured-output capability or less stable instruction following, these constraints are more likely to surface as parsing failures or parameter validation errors, which in turn affects the stability and reproducibility of the evaluation pipeline.

*Table 10.* Distribution of CWE Types in the Benchmark

| CWE ID | Vulnerability Type | Count | Pct. |
|--------|-------------------|-------|------|
| CWE-79 | Cross-site Scripting (XSS) | 25 | 9.7% |
| CWE-74 | Injection | 13 | 5.0% |
| CWE-94 | Code Injection | 12 | 4.7% |
| CWE-22 | Path Traversal | 11 | 4.3% |
| CWE-89 | SQL Injection | 10 | 3.9% |
| CWE-434 | Unrestricted File Upload | 10 | 3.9% |
| CWE-284 | Improper Access Control | 9 | 3.5% |
| CWE-119 | Buffer Errors | 8 | 3.1% |
| CWE-400 | Uncontrolled Resource Consumption | 8 | 3.1% |
| CWE-306 | Missing Authentication | 7 | 2.7% |
| CWE-416 | Use After Free | 7 | 2.7% |
| CWE-20 | Improper Input Validation | 6 | 2.3% |
| CWE-78 | OS Command Injection | 6 | 2.3% |
| CWE-770 | Allocation without Limits | 6 | 2.3% |
| CWE-502 | Deserialization of Untrusted Data | 6 | 2.3% |
| CWE-1321 | Prototype Pollution | 5 | 1.9% |
| CWE-77 | Command Injection | 5 | 1.9% |
| CWE-1333 | Regular Expression DoS (ReDoS) | 5 | 1.9% |
| CWE-125 | Out-of-bounds Read | 5 | 1.9% |
| CWE-703 | Improper Exception Handling | 4 | 1.6% |
| CWE-863 | Incorrect Authorization | 4 | 1.6% |
| CWE-200 | Information Exposure | 4 | 1.6% |
| CWE-23 | Relative Path Traversal | 4 | 1.6% |
| CWE-404 | Improper Resource Shutdown | 3 | 1.2% |
| CWE-73 | External Control of File Name | 3 | 1.2% |
| CWE-918 | Server-Side Request Forgery (SSRF) | 3 | 1.2% |
| CWE-862 | Missing Authorization | 3 | 1.2% |
| CWE-190 | Integer Overflow | 3 | 1.2% |
| CWE-287 | Improper Authentication | 3 | 1.2% |
| CWE-122 | Heap-based Buffer Overflow | 3 | 1.2% |
| CWE-787 | Out-of-bounds Write | 3 | 1.2% |
| CWE-95 | Eval Injection | 2 | 0.8% |
| CWE-124 | Buffer Underflow | 2 | 0.8% |
| CWE-129 | Improper Array Index Validation | 2 | 0.8% |
| CWE-347 | Improper Signature Verification | 2 | 0.8% |
| CWE-843 | Type Confusion | 2 | 0.8% |
| CWE-601 | Open Redirect | 2 | 0.8% |
| CWE-1336 | Server-Side Template Injection (SSTI) | 2 | 0.8% |
| CWE-345 | Insufficient Verification of Authenticity | 2 | 0.8% |
| CWE-134 | Format String Vulnerability | 2 | 0.8% |
| CWE-265 | Privilege Issues | 2 | 0.8% |
| CWE-1050 | Excessive Platform Resource Consumption | 2 | 0.8% |
| CWE-476 | NULL Pointer Dereference | 1 | 0.4% |
| CWE-792 | Incomplete Filtering of Special Elements | 1 | 0.4% |
| CWE-644 | Improper Neutralization of HTTP Headers | 1 | 0.4% |
| CWE-639 | Authorization Bypass (IDOR) | 1 | 0.4% |
| CWE-707 | Improper Neutralization | 1 | 0.4% |
| CWE-126 | Buffer Over-read | 1 | 0.4% |
| CWE-407 | Inefficient Algorithmic Complexity | 1 | 0.4% |
| CWE-303 | Incorrect Implementation of Authentication | 1 | 0.4% |
| CWE-266 | Incorrect Privilege Assignment | 1 | 0.4% |
| CWE-277 | Insecure Inherited Permissions | 1 | 0.4% |
| CWE-1285 | Improper Validation of Quantity in Input | 1 | 0.4% |
| CWE-613 | Insufficient Session Expiration | 1 | 0.4% |
| CWE-27 | Directory Traversal | 1 | 0.4% |
| CWE-755 | Improper Handling of Exceptional Conditions | 1 | 0.4% |
| CWE-123 | Write-what-where Condition | 1 | 0.4% |
| CWE-117 | Log Injection | 1 | 0.4% |
| CWE-352 | Cross-Site Request Forgery (CSRF) | 1 | 0.4% |
| CWE-59 | Symlink Following | 1 | 0.4% |
| CWE-252 | Unchecked Return Value | 1 | 0.4% |
| CWE-833 | Deadlock | 1 | 0.4% |
| CWE-138 | Improper Neutralization of Special Elements | 1 | 0.4% |
| CWE-203 | Observable Discrepancy | 1 | 0.4% |
| CWE-611 | XML External Entity (XXE) | 1 | 0.4% |
| CWE-617 | Reachable Assertion | 1 | 0.4% |
| CWE-348 | Use of Less Trusted Source | 1 | 0.4% |
| CWE-179 | Incorrect Behavior Order | 1 | 0.4% |
| CWE-24 | Absolute Path Traversal | 1 | 0.4% |
| CWE-295 | Improper Certificate Validation | 1 | 0.4% |
| CWE-338 | Weak PRNG | 1 | 0.4% |
| CWE-116 | Improper Encoding/Escaping | 1 | 0.4% |
| CWE-144 | CRLF Injection | 1 | 0.4% |
| CWE-749 | Exposed Dangerous Method | 1 | 0.4% |
| **Total (unique CWEs)** | | **74** | – |

# D. Training Details

We fine-tune Qwen3-32B using full-parameter training on 64 H100 GPUs for 5 epochs. Table 11 summarizes the key hyperparameters.

*Table 11.* Training configuration.

| Hyperparameter | Value |
|---|---|
| Base model | Qwen3-32B |
| Precision | BFloat16 |
| Learning rate | 1e-5 |
| Weight decay | 0.1 |
| Warmup ratio | 0.001 |
| Max sequence length | 65,536 |
| Epochs | 5 |
| Optimizer | AdamW |
| Attention | FlashAttention-2 |
| RoPE scaling | YaRN |
| Parallelism | DeepSpeed ZeRO-3 + SP (size=2) |

# E. Time and Cost Distribution of Agents

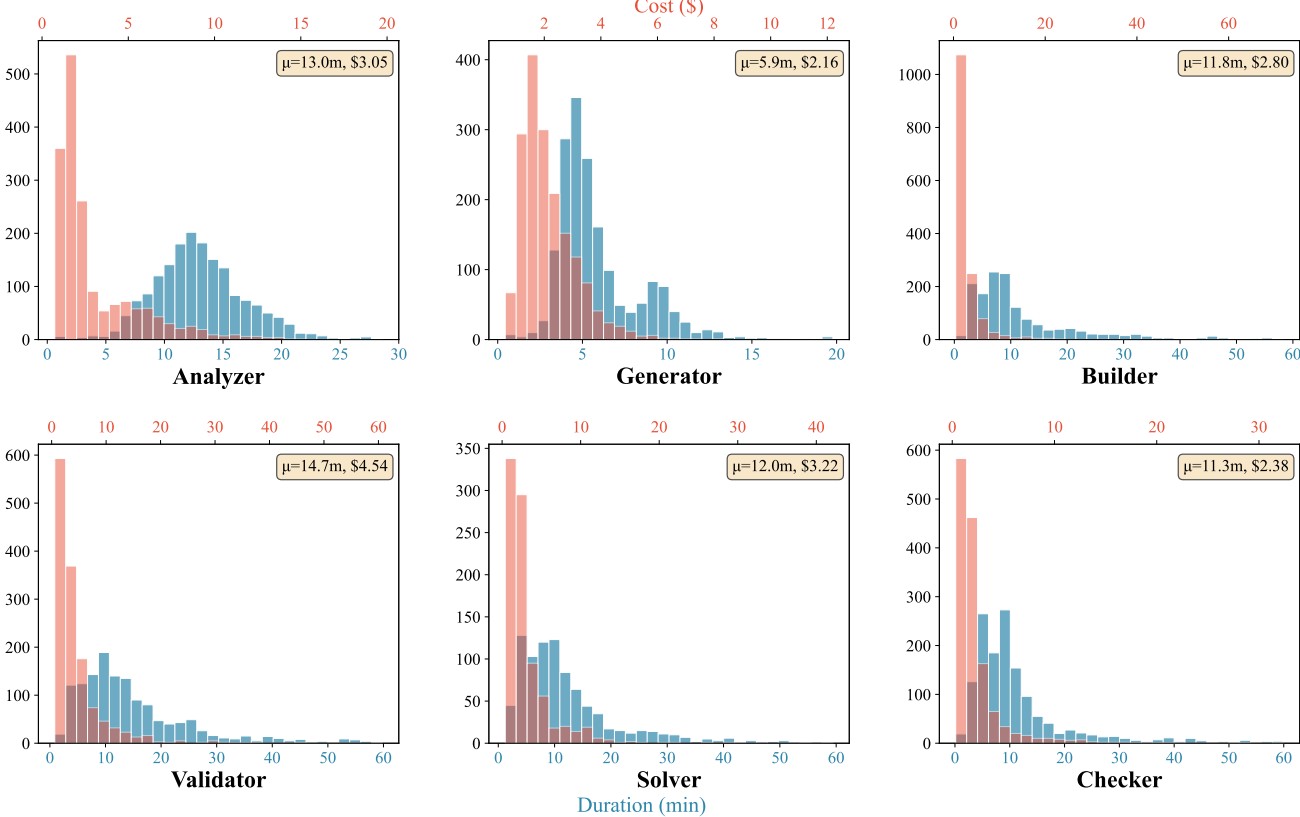

*Figure 6.* Distribution of execution time and cost across six agents.

## F. Comparative Case Study: Manual vs. CVE-Factory (CVE-2021-21384)

CVE-2021-21384 concerns a command injection vulnerability in the `shescape` library caused by improper sanitization of null characters. While the library supports both Unix and Windows platforms, the official (PatchEval) remediation failed to address all affected components. In contrast, CVE-Factory correctly identified the multi-platform scope, generating a more rigorous test suite and a comprehensive patch that addresses the oversight.

In response to the vulnerability report, the maintainers introduced regression tests. However, as illustrated below, the scope of these tests was restricted exclusively to the Unix implementation (`test/unix.test.js`), entirely overlooking the Windows platform. The provided tests were restricted to validating single-quote escaping on Unix, failing to account for vulnerability exploitation vectors on Windows systems.

**PatchEval Test**

```
...
# The implementation focused exclusively on single-quote escaping.
  describe("escape single quotes", function () {
    it("escapes one single quote", function () {
      const input = '' & ls -al';
      const output = escapeShellArg(input);
      assert.strictEqual(output, ''\\'' & ls -al');
    });

    it("escapes two single quotes", function () {
      const input = '' & echo 'Hello world!'';
      const output = escapeShellArg(input);
      assert.strictEqual(output, ''\\'' & echo '\\''Hello world!'\\''');
    });
  });

  describe("null characters", function () {
    const nullChar = String.fromCharCode(0);

    it("removes one null character", function () {
      const input = 'foo' && ls${nullChar} -al ; echo 'bar';
      const output = escapeShellArg(input);
      assert.strictEqual(output, 'foo'\\'' && ls -al ; echo '\\''bar');
    });

    it("removes multiple null character", function () {
      const input = 'foo'${nullChar}&&ls -al${nullChar};echo 'bar';
      const output = escapeShellArg(input);
      assert.strictEqual(output, 'foo'\\''&&ls -al;echo '\\''bar');
    });
  });
});
```

Relying on this limited test suite, the maintainers released the following solution. Although it correctly sanitized `src/unix.js`, it failed to apply the corresponding logic to `src/win.js`, leaving the Windows implementation vulnerable to the same attack vector.

**PatchEval Solution**

```
cat <<'EOF' > /workspace/fix.patch
diff --git a/src/unix.js b/src/unix.js
index 89fc3bdc4..d0671b546 100644
```

```
--- a/src/unix.js
+++ b/src/unix.js
@@ -11,7 +11,7 @@
  * @returns {string} The escaped argument.
  */
 function escapeShellArg(arg) {
-  return arg.replace(/'/g, '\\'');
+  return arg.replace(/\u{0}/gu, "").replace(/'/g, '\\'');
 }

 module.exports.escapeShellArg = escapeShellArg;

EOF

cd /workspace/shescape
git apply --whitespace=nowarn  /workspace/fix.patch
```

Conversely, CVE-Factory analyzed the repository structure and generated a comprehensive test suite. As shown below, the generated tests extend coverage beyond Unix to include the Windows environment, specifically verifying the correctness of double-quote escaping on Windows platforms.

**CVE-Factory Test**

```python
import subprocess
import json
import pytest

REPO_PATH = "/workspace/shescape"

class TestUnixNullCharacterHandling:
    """Test that Unix escaping properly removes null characters."""

    def test_single_null_character_removed(self):
        """Test that a single null character is removed from input."""
        test_code = """
        const {escapeShellArg} = require('./src/unix');
        const nullChar = String.fromCharCode(0);
        const input = "test" + nullChar + "data";
        const result = escapeShellArg(input);
        console.log(JSON.stringify({
            input: input,
            output: result,
            inputLength: input.length,
            outputLength: result.length,
            hasNull: result.includes(nullChar)
        }));
        """
        result = subprocess.run(
            ['node', '-e', test_code],
            cwd=REPO_PATH,
            capture_output=True,
            text=True
        )
```

```python
        assert result.returncode == 0, f"Node.js error: {result.stderr}"
        data = json.loads(result.stdout)

        # After fix: null character should be removed
        assert data['hasNull'] == False, "Null character was not removed"
        assert '\x00' not in data['output'], "Null character found in output"

    # ...

    def test_null_with_quotes_properly_escaped(self):
        """Test complex payload with null characters and quotes."""
        test_code = """
        const {escapeShellArg} = require('./src/unix');
        const nullChar = String.fromCharCode(0);
        const input = "foo' && ls" + nullChar + " -al ; echo 'bar";
        const result = escapeShellArg(input);
        const expected = "foo'\\\\'' && ls -al ; echo '\\\\''bar";
        console.log(JSON.stringify({
            output: result,
            expected: expected,
            hasNull: result.includes(nullChar),
            matches: result === expected
        }));
        """
        result = subprocess.run(
            ['node', '-e', test_code],
            cwd=REPO_PATH,
            capture_output=True,
            text=True
        )
        assert result.returncode == 0, f"Node.js error: {result.stderr}"
        data = json.loads(result.stdout)

        # Null character should be removed and quotes should be escaped
        assert data['hasNull'] == False, "Null character was not removed"
        assert data['matches'], f"Expected '{data['expected']}', \
                                 got '{data['output']}'"

    # ...

class TestWindowsNullCharacterHandling:
    """Test that Windows escaping properly removes null characters."""

    def test_single_null_character_removed_windows(self):
        """Test that a single null character is removed on Windows."""
        test_code = """
        const {escapeShellArg} = require('./src/win');
        const nullChar = String.fromCharCode(0);
        const input = "test" + nullChar + "data";
        const result = escapeShellArg(input);
        console.log(JSON.stringify({
            output: result,
```

```
                hasNull: result.includes(nullChar)
            }));
            """
        result = subprocess.run(
            ['node', '-e', test_code],
            cwd=REPO_PATH,
            capture_output=True,
            text=True
        )
        assert result.returncode == 0, f"Node.js error: {result.stderr}"
        data = json.loads(result.stdout)

        assert data['hasNull'] == False,\
            "Null character was not removed on Windows"
        assert data['output'] == "testdata"

    # ...

    def test_null_with_double_quotes_windows(self):
        """Test complex payload with null characters on Windows."""
        test_code = """
        const {escapeShellArg} = require('./src/win');
        const nullChar = String.fromCharCode(0);
        const input = 'foo" && ls' + nullChar + ' -al ; echo "bar';
        const result = escapeShellArg(input);
        const expected = 'foo"" && ls -al ; echo ""bar';
        console.log(JSON.stringify({
            output: result,
            expected: expected,
            hasNull: result.includes(nullChar),
            matches: result === expected
        }));
        """
        result = subprocess.run(
            ['node', '-e', test_code],
            cwd=REPO_PATH,
            capture_output=True,
            text=True
        )
        assert result.returncode == 0, f"Node.js error: {result.stderr}"
        data = json.loads(result.stdout)

        assert data['hasNull'] == False
        assert data['matches'], \
            f"Expected '{data['expected']}', got '{data['output']}'"

# ... (TestMainAPINullCharacterHandling omitted) ...

if __name__ == "__main__":
    pytest.main([__file__, "-v"])
```

We validated the PatchEval patch against the CVE-Factory test suite. The execution logs below corroborate the oversight: while Unix-specific tests passed, the Windows-targeted tests failed immediately, exposing the persisting vulnerability.

---

**Test Log (PatchEval Solution + CVE-Factory Test)**

```
=========================== PASSES ============================
================= short test summary info ====================
...
test_vuln.py::TestUnixNullCharacterHandling::
test_only_null_character
PASSED test_vuln.py::TestMainAPINullCharacterHandling::
test_escape_removes_null
PASSED test_vuln.py::TestMainAPINullCharacterHandling::
test_quote_removes_null
PASSED test_vuln.py::TestMainAPINullCharacterHandling::
test_escape_all_removes_null
PASSED test_vuln.py::TestMainAPINullCharacterHandling::
test_command_injection_prevented
FAILED test_vuln.py::TestWindowsNullCharacterHandling::
test_single_null_character_removed_windows – AssertionError:
Null character was not removed on Windows
FAILED test_vuln.py::TestWindowsNullCharacterHandling::
test_multiple_null_characters_removed_windows – AssertionError:
Null characters were not removed on Windows
FAILED test_vuln.py::TestWindowsNullCharacterHandling::
test_null_with_double_quotes_windows – AssertionError:
Null character was not removed on Windows
============== 3 failed, 21 passed in 0.65s ==============
```

CVE-Factory generated a robust solution that enforces consistency across platforms. As demonstrated in the script below, the tool applies the necessary null character sanitization to both Unix and Windows source files, effectively mitigating the vulnerability across the entire attack surface.

---

**CVE-Factory Solution**

```bash
c#!/bin/bash
set –e

# Solution script for shescape null character bypass vulnerability
# Applies the fix to both src/unix.js and src/win.js

cd /workspace/shescape

# Fix src/unix.js – add null character removal before quote escaping
cat > /workspace/shescape/src/unix.js << 'EOF'
function escapeShellArg(arg) {
  return arg.replace(/\u{0}/gu, "").replace(/'/g, '\\'');
}

module.exports = {
  escapeShellArg,
};
EOF

# Fix src/win.js – add null character removal before quote escaping
cat > /workspace/shescape/src/win.js << 'EOF'
```

```
function escapeShellArg(arg) {
  return arg.replace(/\u{0}/gu, "").replace(/"/g, `""`);
}

module.exports = {
  escapeShellArg,
};
EOF

echo "Fix applied: Added null character removal to escapeShellArg functions"
```

Finally, the validation logs confirm that the solution generated by CVE-Factory successfully passes the comprehensive test suite. This confirms that the tests target a genuine vulnerability that requires remediation, which CVE-Factory successfully addressed.

**Test Log (CVE-Factory Solution + CVE-Factory Test)**

```
============================== PASSES ==============================
================== short test summary info ==================
...
test_vuln.py::TestUnixNullCharacterHandling::
test_only_null_character
PASSED test_vuln.py::TestMainAPINullCharacterHandling::
test_escape_removes_null
PASSED test_vuln.py::TestMainAPINullCharacterHandling::
test_quote_removes_null
PASSED test_vuln.py::TestMainAPINullCharacterHandling::
test_escape_all_removes_null
PASSED test_vuln.py::TestMainAPINullCharacterHandling::
test_command_injection_prevented
FAILED test_vuln.py::TestWindowsNullCharacterHandling::
test_single_null_character_removed_windows – AssertionError:
Null character was not removed on Windows
PASSED test_vuln.py::TestWindowsNullCharacterHandling::
test_multiple_null_characters_removed_windows – AssertionError:
Null characters were not removed on Windows
PASSED test_vuln.py::TestWindowsNullCharacterHandling::
test_null_with_double_quotes_windows – AssertionError:
Null character was not removed on Windows
============== 0 failed, 24 passed in 1.13s ==============
```

In conclusion, the tests and solution generated by CVE-Factory for this CVE prove to be significantly better than the official baseline (PatchEval).

# G. Limitations

Our study focuses on establishing the feasibility and utility of scalable CVE task synthesis, but several directions remain underexplored due to resource and infrastructure constraints. First, our training experiments are limited to Qwen3-32B and supervised fine-tuning on distilled trajectories. Although this already yields substantial gains, we have not evaluated whether larger open-weight models or stronger proprietary models would exhibit more favorable scaling behavior when trained on CVE-Factory tasks. Second, due to the high cost of maintaining thousands of interactive vulnerability environments, we do not perform reinforcement learning directly in executable CVE environments. Such online training could further improve exploration, debugging, and test-driven repair, especially for long-horizon failures where supervised trajectories provide limited feedback. Third, while CVE-Factory achieves strong automation, its current pipeline is still optimized for publicly available, patched, and containerizable CVEs. Vulnerabilities involving proprietary software, hardware dependencies, GUI-only workflows, or complex distributed deployments remain challenging.

