# OpenReview forum: "CVE-Factory: Scaling Expert-Level Agentic Tasks for Code Security Vulnerability"
_ICML.cc/2026/Conference — ICML 2026 spotlight_

### Official Review · Reviewer_7RLH · 2026-03-06

**Soundness:** 4
**Presentation:** 4
**Significance:** 4
**Originality:** 3
**Overall Recommendation:** 5
**Confidence:** 4

**Summary:**

The authors propose CVE-Factory, a multi-agent framework designed to autonomously transform sparse CVE metadata into fully executable agentic environments. To handle the massive context window required for full CVE reproduction, the framework employs a decouple-couple architecture that distributes the workload across specialized agents operating under a centralized Orchestrator. The authors demonstrate that CVE-Factory matches human expert reproductions in fidelity and correctness. The downstream contributions include the LiveCVEBench benchmark and over 1,000 training environments. Fine-tuning Qwen3-32B on these generated trajectories yields performance improvements on LiveCVEBench and Terminal Bench.

**Compliance With Llm Reviewing Policy:**

Affirmed.

**Final Justification:**

The authors provided further insights and addressed my questions, so the positive score is kept.

**Key Questions For Authors:**

- Given the agents' tendency to override explicit system constraints. What can be the potential solutions to enforce these constraints during generation?
- The paper highlights that 27.75% of LiveCVEBench tasks require multi-container orchestration. Does the multi-agent framework's performance degrade, and by how much, as the architectural complexity of the target repository increases?

**Limitations:**

yes

**Strengths And Weaknesses:**

Strengths:
- The paper makes a significant contribution to the landscape of vulnerability data curation, as it often poses a great challenge for researchers to collect abundant data. By automating the translation of static CVE metadata into these complex environments, CVE-Factory directly addresses the severe scalability bottleneck of expert-driven reproduction. In addition, the inclusion of the high-fidelity training environments and a benchmark provides resources for the community.
- The multi-agent framework is a carefully designed pipeline that fully utilizes the coding capability of LLMs. Instead of relying on simple heuristics and prompts, the decoupling and coupling approach defines the scope and task for the agents. Building environments is also crucial for the verification process to reproduce the intended vulnerabilities.
- The quality of the reproduction is impressive, with faithful reconstruction of the quality of the human.

Weaknesses:
- The presentation of the results occasionally lacks a deep discussion on why certain phenomena occur. For example, the paper notes a severe drop in performance for post-release CVEs across almost all models, suggesting potential data contamination in existing benchmarks. A more rigorous analysis of these agent trajectories would be beneficial to allow a better understanding of zero-shot-like agentic reasoning flaws.

---

> ### Author Rebuttal · Authors · 2026-03-29
>
> 1. W1 (**Performance drop on post-release CVEs**): We deeply appreciate this constructive suggestion! To understand performance drop on post-release CVEs, we conducted a rigorous trajectory analysis. The performance drop is primarily driven by implicit data contamination. **While LLMs might not directly memorize the exact CVE patches, they possess a significantly higher overall familiarity with the pre-release codebases and frameworks, enabling them to locate vulnerabilities much faster.** The "first successful edit step" (modifying code without syntax errors) is noticeably delayed in post-release settings: from 6.4 to 10.3 for Qwen3-Coder, and 5.4 to 6.9 for Claude-Sonnet-4.5. On the pre-release CVE-2025-10824 , GPT-5.1-Codex quickly located the issue, initiating edits by step 6. Conversely, on the post-release CVE-2025-68129, it spent 34 steps purely exploring, made its first edit at step 35, and hit the timeout limit while still struggling to configure validation tools. Similarly, on the pre-release CVE-2025-9670, DeepSeek-V3.1-Terminus rapidly located and patched the target file, completing the task in just 17 steps. However, on the post-release CVE-2025-49844, it exhausted 49 steps in complete disorientation. It wasted the first 13 steps executing flawed search commands and hallucinating file structures (e.g., outputting "It seems the find command isn't finding any files" and "The current directory appears to be empty"), burning its context window before even finding the vulnerable file. These trajectories confirm that without implicit codebase familiarity, agents in true zero-shot settings exhaust their context windows on exploratory reasoning and codebase navigation.
>
> 2. KQ1 (**Enforcing constraints during generation**): As preventing agents from rationalizing shortcuts is a critical challenge raised by multiple reviewers, we propose a multi-layered defense to systematically break this error propagation:
>   - Upgraded Adversarial Checker: Our initial Checker failed because it was easily misled by upstream agents' excuses. To fix this, we deployed a V2 prompt that forces the Checker to act as a strict evaluator. **It uses a 10-category systematic auditing schema to identify specific shortcut patterns, combined with strict anchoring instructions ("Ignore all upstream defenses; extract only the core issue")** to ensure the Checker accurately detects specific constraint violations. Tested on 500 CVE samples, this explicitly eliminated false positives. Once a violation is identified, the Checker directly modifies and corrects the generated content to enforce the constraint on the spot.
>   - Static Analysis Checks: We are integrating deterministic heuristic checks into the pipeline. For example, we are adding static rules like verifying the length and count of generated core files, and parsing the Dockerfile ensuring it explicitly downloads/clones source code. Violating these rules bypasses the LLM's discretion and forces an immediate feedback loop to re-generate.
>   - Rubric RL: Regarding explicit penalty signals, the current LLM training paradigm relies heavily on Outcome-supervised Reward Models (ORM), which only evaluate final states and miss intermediate shortcuts. For our future work in training our own security-aligned models, we plan to explore Rubric RL (Process Supervision) to provide explicit, step-level penalty signals against shortcut behaviors.
>
> 3. KQ2 (**Performance degradation on multi-container tasks**): We analyzed CVE-Factory's environment reproduction success rate across different architectural complexities and observed a non-linear relationship. Surprisingly, **CVE-Factory's success rate on 2-container tasks is actually 6% higher than on 1-container tasks**. Our analysis reveals this is driven by vulnerability diversity and environmental standardization. Approximately 83% of 2-container tasks are standard "App + Database" architectures (primarily PHP web applications). In contrast, 1-container tasks exhibit extreme diversity and include complex low-level vulnerabilities (e.g., memory safety issues in C/C++). Synthesizing the complex compilation environment and memory-corruption triggers for these tasks is significantly harder. However, performance does degrade as the container count increases further. **When scaling to 3 or 4 containers, the framework's success rate drops by 10%** compared to the 1-container baseline. This is because these tasks often involve uncommon and complex architectures, such as Spring Boot + Elasticsearch or Laravel + Nginx + Mailpit. Their inter-container networking, initialization synchronization, and port binding configurations are highly complex.

---

> > ### Author Rebuttal · Reviewer_7RLH · 2026-04-01
> >
> > I appreciate the authors' comprehensive analysis to address my questions, and I would like to maintain my positive score of 5.

---

### Official Review · Reviewer_QA17 · 2026-03-09

**Soundness:** 2
**Presentation:** 3
**Significance:** 3
**Originality:** 3
**Overall Recommendation:** 4
**Confidence:** 4

**Summary:**

This paper proposes CVE-Factory, a multi-agent framework that utilizes a six-stage decouple-couple architecture to automatically transform sparse CVE metadata into fully executable agentic tasks. This paper also constructs LiveCVEBench, a continuously updated benchmark designed for evaluating vulnerability reproduction. Experimental results demonstrate that CVE-Factory achieves expert-level reproduction quality.

**Compliance With Llm Reviewing Policy:**

Affirmed.

**Final Justification:**

The authors' response has partially addressed my questions. Overall, I am willing to keep my original positive score. However, the risk of superficial fixes and how to mitigate error propagation still need to be addressed in future work.

**Key Questions For Authors:**

1. I am interested in understanding how much each component contributes to the final performance.

2. Is there a risk that relying solely on test scripts for evaluation might lead to fixes that pass the tests but fail to address the underlying root cause?

**Limitations:**

Yes.

**Strengths And Weaknesses:**

Strengths:

1. This paper proposes CVE-Factory , a novel multi-agent framework with a decouple-couple architecture to automate complex CVE reproduction.

2. This paper construct LiveCVEBench, a continuously updated benchmark that captures emerging real-world vulnerability distribution shifts.

3. The work contributes a large-scale synthesis of code security tasks, generating executable environments that serve as a crucial resource for downstream LLM fine-tuning.

Weaknesses:

1. The paper would benefit significantly from empirical comparisons with existing automated baselines. Including prior works such as CVE-Genie, as well as simpler setups like a naive single-agent or ReAct configuration, would greatly strengthen the assessment of the proposed six-stage pipeline.

2. Soundness. The evaluation of the Decouple-Couple architecture would be significantly strengthened by including architectural ablation studies. While the current results demonstrate balanced execution time across agents, additional empirical evidence is needed to directly correlate this balance with architectural superiority or improved success rates.

3. Soundness. There is a potential concern that exposing dynamically generated test scripts to the evaluating agents may inadvertently encourage the generation of plausible but incorrect ad-hoc patches, rather than addressing the root cause. This possibility is suggested by the 'Invalid Solution' instances in Table 6, which passed automated tests but did not withstand human inspection.

Other Comments:

1. While the framework thoughtfully incorporates feedback mechanisms to address early-stage errors (such as mock implementations), empirical observations suggest that these measures may not be fully sufficient in practice. It appears that downstream agents might inadvertently prioritize task completion over strict adherence to constraints, leading them to unintentionally inherit and rationalize upstream hallucinations rather than actively correcting them. Further analysis or enhanced strategies to break this chain of error propagation would significantly strengthen the robustness of the proposed system.

2. Typo in Section 4.2.3: On page 7, the text currently states, 'Figure 6 reveals a concerning pattern. We partition LiveCVEBench by each model's release date...' However, based on the context and data presented, it appears that this reference may be intended to cite Figure 4 instead.

3. I commend the authors for their transparency in Section 4.2.2 regarding the 6 failures attributed to external factors (e.g., network timeouts). To further strengthen the scientific rigor of the evaluation, it would be valuable to conduct a targeted re-run of these specific 6 cases. Isolating these instances from transient environmental issues would allow us to ascertain their true outcomes and provide a clearer picture of the framework's inherent performance limits.

---

> ### Author Rebuttal · Authors · 2026-03-29
>
> 1. W1 & W2 & KQ1 (**More Baseline & Architecture Ablation**): Per your excellent suggestion, we conducted comparative and ablation studies on a random sample of 31 recent CVEs (2025), which were reproduced and released by CVE-Genie. The results are summarized below:
> ||Success|
> |---|---|
> |Claude Code (Single-agent)|0|
> |CVE-Genie|48.4|
> |CVE-Factory|**90.3**|
> |*Ablation*||
> |Merge Stage 1-3|71.0|
> |Merge Stage 4-6|58.1|
>
>     As shown in the table, we employ Claude Code (SOTA Agent) as our single-agent baseline, which completely fails (0%) due to the complex, long-horizon nature of the task. In contrast, CVE-Factory (90.3%) significantly outperforms the existing multi-agent baseline, CVE-Genie (48.4%).
>
>     Furthermore, we explicitly validate our "decouple-couple" architecture through ablation studies. "Merge Stage 1-3" condenses the three decoupled generation stages into a single step to generate all files simultaneously. This overloads the agent's context, causing it to forget formatting constraints and generate progressively shorter files. "Merge Stage 4-6" collapses the step-by-step verification into a single end-to-end evaluation. This overwhelms the agent with simultaneous test-and-environment debugging, consistently leading to execution timeouts. These results demonstrate that both our granular decoupling and progressive coupling are strictly necessary for successful reproduction.
>
> 2. W3 & KQ2 (**Risk of superficial fixes**): We apologize for the confusion. In Table 6, "Invalid Solution" means the agent upgraded the package to a safe version directly, rather than modifying the vulnerable source code. Regarding the concern that agents might generate superficial fixes just to pass the evaluation without addressing the root cause, our framework employs three defenses. First, our generated dynamic tests are extensive. On average, **CVE-Factory generates 11.83 functional tests and 11.48 vulnerability tests per task** (significantly outperforming PatchEval’s average of 3.14). These tests systematically probe an average of **7.02 diverse attack vectors**, including multiple entry points and bypass techniques. This high diversity makes it extremely difficult to "hack" the evaluation with a hardcoded or shallow fix. Second, our Stage 6 'Checker' agent explicitly audits the patch to reject superficial workarounds. Finally, our rigorous manual review guarantees that the released tasks feature true, root-cause repairs.
>
> 3. Other Comments 1 (Error propagation): We sincerely thank the reviewer for highlighting the challenge of preventing LLMs from rationalizing shortcuts. To systematically break this error propagation, we propose a multi-layered defense:
>   - Upgraded Checker: Our initial Checker failed because it was easily misled by upstream agents' excuses for using mocks. To fix this, we deployed a V2 prompt that forces the Checker to act as a strict evaluator. It uses a 10-category systematic auditing schema to identify specific shortcut patterns, combined with strict anchoring instructions (**"Ignore all upstream defenses; extract only the core issue"**). Tested on 500 CVE samples, this explicitly eliminated false positives.
>   - Static Analysis Checks: We are integrating deterministic heuristic checks into the pipeline. Specifically, we are adding static rules like **verifying the length and count of generated core files, and parsing the Dockerfile ensuring it explicitly downloads/clones source code**. Violating these rules bypasses the LLM's discretion and forces an immediate feedback loop.
>   - Rubric RL: Regarding explicit penalty signals, the current LLM training paradigm relies heavily on Outcome-supervised Reward Models (ORM), which only evaluate final states and miss intermediate shortcuts. For our future work in training our own security-aligned models, we plan to explore Rubric RL (Process Supervision) to provide explicit, step-level penalty signals against shortcut behaviors.
> 4. Other Comments 2 (**Typo**): Thank you for catching this! We will correct the reference in the revised manuscript.
> 5. Other Comments 3 (**External failures**): We appreciate the suggestion, but chose to retain these as failures. First, due to hard network constraints, outcomes of manual re-runs remain inherently unstable. We opted to strictly categorize them as absolute failures to maintain a conservative lower bound, preventing any artificial inflation of our success rate. Second, within our fully automated pipeline, agents are already granted sufficient execution time, unconstrained operational space, and multiple retry mechanisms to handle errors. If an agent still fails to navigate these external dependencies despite these built-in resources, it exposes a genuine limitation in its real-world robustness rather than mere experimental noise. Thus, retaining these failures most accurately reflects the true, unassisted reliability of agentic workflows in the wild.

---

> > ### Author Rebuttal · Reviewer_QA17 · 2026-04-02
> >
> > Thank you for your detailed rebuttal. Some of my concerns have been nicely addressed. The risk of superficial fixes may indeed ultimately rely on manual auditing instead of automated approaches. I also appreciate the authors sharing several promising directions that could potentially mitigate error propagation in future work, though their effectiveness and ability to fundamentally resolve this issue remain unclear at this stage. I believe my original score for the paper is reasonable and will thus maintain it.

---

### Official Review · Reviewer_e8nF · 2026-03-10

**Soundness:** 3
**Presentation:** 3
**Significance:** 3
**Originality:** 3
**Overall Recommendation:** 5
**Confidence:** 3

**Summary:**

CVE-Factory is a multi-agent framework that automatically transforms sparse CVE metadata into fully executable vulnerability repair tasks (environment + tests + solution). It addresses the core bottleneck that manual CVE reproduction costs experts 10+ hours each and cannot scale. The pipeline has six stages: three decoupled generation stages (information collection, file synthesis, Docker environment construction) followed by three progressive verification stages that align the components. A central Orchestrator manages agent activation and routes failures back to responsible creators via a file-ownership feedback mechanism. Validated against PatchEval, CVE-Factory achieves 95% solution correctness and 96% environment fidelity at 6–30x speedup. Applied to 554 recent real-world CVEs, it achieves 66.2% verified success. The authors further release LiveCVEBench (190 tasks, 14 languages, continuously updated) and show that fine-tuning Qwen3-32B on 4k synthesized trajectories improves it from 5.3% to 35.8% on LiveCVEBench, surpassing Claude Sonnet 4.5, with gains transferring to Terminal Bench.

**Compliance With Llm Reviewing Policy:**

Affirmed.

**Key Questions For Authors:**

1. The paper acknowledges that when multiple valid fix strategies exist, CVE-Factory tests may implicitly expect a specific post-patch behavior, causing expert solutions to fail. How many of the reported 96% environment fidelity cases involved unambiguous single-strategy fixes versus cases where Ct was inadvertently written around Ce's approach? A stratified breakdown would significantly strengthen this claim.

2. The current approach—prompting constraints plus a Checker—demonstrably fails. Have the authors explored static analysis to detect mock patterns before stage completion, circuit-breaker heuristics when Analyzer scopes down reproduction, or explicit penalty signals during agent training? This is arguably the most important engineering challenge for practical deployment.

3. CVE-Factory targets CVEs from May–December 2025, some of which may remain unpatched in production systems. The Impact Statement dismisses societal concerns without substantive discussion. Does the released training data include working PoCs for unpatched vulnerabilities? What coordinated disclosure protocols were followed? This is a non-trivial concern given the paper's stated goal of making exploit reproduction more scalable and accessible.

**Limitations:**

see weaknesses and questions

**Strengths And Weaknesses:**

**Strengths**

- The decouple-couple design cleanly solves the context overflow problem that makes single-agent CVE reproduction intractable. The file-ownership feedback mechanism—routing failures back to the original creator agent rather than restarting—is a practical and non-obvious engineering contribution.
- Training only on Python/JS/Go yet achieving 867% improvement on PHP, and transferring to Terminal Bench's non-security tasks, suggests the synthesized trajectories teach broadly applicable agentic reasoning skills rather than domain-specific memorization.
- The finding that nearly all evaluated models degrade on post-release CVEs is a valuable empirical result motivating LiveCVEBench's continuous update design, and raises important questions about contamination in existing static benchmarks.
- The paper quantifies failure modes in detail (mock: 35.9%, static tests: 35.2%, etc.) and candidly diagnoses the root cause—agents prioritizing task completion over constraints—rather than glossing over limitations.

**Weaknesses**

- Test quality is assessed by an LLM-as-judge, introducing circular reasoning: one LLM evaluating another's blind spots. There is no coverage metric, mutation testing, or attack-path enumeration. The CVE-2021-21384 case is illustrative but anecdotal—it doesn't establish that such oversights are systematically caught.
- Despite explicit constraints and a dedicated Checker, 52 mock and 51 static test failures persist. The Discussion correctly identifies this as a fundamental tension—agents rationally fall back when direct approaches fail—but does not propose a path forward.

---

> ### Author Rebuttal · Authors · 2026-03-29
>
> 1. W1(**Test quality and coverage**): We respectfully clarify a misunderstanding regarding "circular reasoning." We do not rely solely on an LLM-as-judge for final test quality. While agents assist in localization and analysis, **all outcomes and test validations are strictly reviewed by humans**. Regarding test metrics, our generated tests are primarily end-to-end, network-based exploits (e.g., HTTP requests targeting a black-box Docker container). Conventional tools like pytest-cov cannot easily measure the coverage of backend code for such out-of-process interactions. However, on the subset of tasks where backend coverage is measurable, empirical data shows CVE-Factory systematically outperforms the human-expert baseline (PatchEval) in 83.3% of comparable tasks. Specifically, **CVE-Factory achieves an average coverage of 59.00%, compared to PatchEval's 43.57%**. Furthermore, addressing your point on attack-path enumeration, when isolating vulnerability-specific exploits (test_vuln), CVE-Factory achieves 44.00% coverage—exactly double the human baseline (21.96%). This 2x deep-path coverage, combined with an average of **7.02 diverse attack vectors per task**, quantitatively proves that our tests are significantly more comprehensive, probing deeply into vulnerable logic rather than superficial endpoints.
> 2. W2 & KQ2 (**Preventing mock fallbacks**): We sincerely thank the reviewer for highlighting the challenge of preventing LLMs from rationalizing shortcuts. To systematically break this error propagation, we propose a multi-layered defense:
>   - Upgraded Checker: Our initial Checker failed because it was easily misled by upstream agents' excuses for using mocks. To fix this, we deployed a V2 prompt that forces the Checker to act as a strict evaluator. It uses a 10-category systematic auditing schema to identify specific shortcut patterns, combined with strict anchoring instructions (**"Ignore all upstream defenses; extract only the core issue"**). Tested on 500 CVE samples, this explicitly eliminated false positives.
>   - Static Analysis Checks: We greatly appreciate your suggestion of heuristic checks, and we are currently incorporating them into our pipeline. Specifically, we are adding static rules like **verifying the length and count of generated core files, and parsing the Dockerfile ensuring it explicitly downloads/clones source code**. Violating these rules bypasses the LLM's discretion and forces an immediate feedback loop.
>   - Rubric RL: Regarding explicit penalty signals, the current LLM training paradigm relies heavily on Outcome-supervised Reward Models (ORM), which only evaluate final states and miss intermediate shortcuts. For our future work in training our own security-aligned models, we plan to explore Rubric RL (Process Supervision) to provide explicit, step-level penalty signals against shortcut behaviors.
> 3. KQ1(**Multi-strategy fixes**): To investigate whether our generated tests ($C_t$) inadvertently overfit to a specific implementation ($C_e$), we conducted a stratified analysis to evaluate their compatibility with diverse, valid fix strategies. The results demonstrate that $C_t$ is actually more robust to multi-strategy fixes than human-expert tests (PatchEval). Specifically, **CVE-Factory accommodates multiple valid fix strategies in 72.5% of cases**. In contrast, **PatchEval accommodates diverse fixes in only 55.0%**. Only 18.8% of cases were strictly single-strategy. This breakdown proves our 96% fidelity is not driven by test overfitting. Instead, CVE-Factory generates robust, end-to-end behavioral tests that validate security properties implementation-agnostically, outperforming human baselines.
> 4. KQ3(**Security ethics**):All our evaluated CVEs are sourced from the public CVElistV5 database, and we strictly select only vulnerabilities that already have official patches. We did not discover or disclose any new vulnerabilities. For our open-source release, we enforce three safeguards: (1) explicitly excluding any CVEs that remain unpatched; (2) applying desensitization techniques to high-risk PoCs to prevent weaponization; and (3) releasing the data under a research-only license to strictly limit the scope of usage. We will expand our Impact Statement in the revision to include a detailed discussion of these ethical considerations.

---

> > ### Author Rebuttal · Reviewer_e8nF · 2026-04-01
> >
> > Thank you for your response. My primary concern has been adequately addressed, and I therefore maintain my original assessment (5: Accept).

---

### Official Review · Reviewer_d3Ky · 2026-03-13

**Soundness:** 3
**Presentation:** 3
**Significance:** 3
**Originality:** 3
**Overall Recommendation:** 4
**Confidence:** 4

**Summary:**

This paper presents CVE-Factory, a multi-agent framework that automatically transforms sparse CVE metadata into fully executable agentic task packages (including environments, tests, and solutions). The core design is a six-stage "decouple-couple" pipeline, where six specialized agents collaborate under a central Orchestrator. The paper makes three contributions: (1) cross-validation against human expert reproductions from PatchEval demonstrates expert-level quality (95% solution correctness, 96% environment fidelity); (2) LiveCVEBench, a continuously updated benchmark of 190 tasks spanning 14 languages; and (3) synthesis of 1,000+ training tasks, where fine-tuned Qwen3-32B improves from 5.3% to 35.8% on LiveCVEBench, approaching Claude Sonnet 4.5, with gains generalizing to Terminal Bench.

**Compliance With Llm Reviewing Policy:**

Affirmed.

**Key Questions For Authors:**

1. Is there any solution to reduce the false positives? Given we still have a not small number of false positives. For the later scaling, it is hard to filter by human effort.

**Limitations:**

1. The false positive problem in the real-world experiment deserves more attention. Out of 499 reported successes, 187 failed manual verification — a 37.5% false positive rate. While the paper analyzes failure causes, this means that without human verification, over one-third of "successful" tasks are actually flawed. The "expert-level quality" claim is primarily supported by PatchEval cross-validation (where CVEs are relatively well-documented), but quality degrades noticeably on real-world long-tail distributions. The paper should more clearly discuss under what conditions CVE-Factory's output can be trusted without manual review.
2. Missing ablation experiments to validate individual design choices. Maybe we need a very basic single agent for CVE reproducting as the baseline for comparing our current multi-agent design's correctness.

**Strengths And Weaknesses:**

1. Meaningful aspect. CVE reproduction is a well-known bottleneck in security research — experts spend 10+ hours per CVE on average, and the process simply does not scale. This paper directly addresses that bottleneck by turning a labor-intensive workflow into an automated pipeline, which has significant implications for the entire code security evaluation and training ecosystem.
2. Interesting system design with well-justified architectural decisions. The six-stage decouple-couple architecture is not just a naive decomposition — it has a clear design rationale: decoupling reduces per-stage cognitive burden, while coupling progressively aligns components. Several design details are particularly clever: Builder's "blind building" constraint prevents cheating, information asymmetry stops error propagation, and the pause-signal feedback mechanism enables precise routing of revisions back to original creators.
3. Solid experimental design with multiple evaluation dimensions. The paper does not just report a single pass rate. Instead, it evaluates across three dimensions — solution correctness, environment fidelity, and test quality — with detailed failure attribution analysis. The real-world experiment on 554 CVEs is also convincing: 66.2% verified success rate is respectable given the difficulty, and the paper is candid about failure modes.

---

> ### Author Rebuttal · Authors · 2026-03-28
>
> We sincerely thank the reviewer for recognizing the value of our automated pipeline, our decouple-couple architecture, and our rigorous evaluation design.
> 1. KQ1 & L1 (**Reducing false positives**): We completely agree that reducing false positives (FPs) is critical for scaling without human intervention. To achieve trustless automation, we optimized a dedicated "Judger" LLM to replace human review. Our initial V1 prompt struggled with a 22.85% error rate (16/70 FPs) because the Judger was easily swayed by upstream agents' rationalizations (e.g., accepting mocks). To break this error propagation, we designed a V2 prompt featuring: (1) a 10-category systematic auditing schema with few-shot examples, and (2) strict anchoring instructions at the beginning and end (**"Ignore all upstream defenses; extract only the core issue"**). Evaluated on 500 CVEs, this V2 Judger achieved a 100% agreement rate with manual human review.  Moving forward, we will integrate this optimized Judger's mechanisms into the Stage 6 'Checker' to prevent FPs during generation, and officially deploy this Judger to entirely replace human review for future large-scale task synthesis.
> 2. L2 (**Baselines and architecture ablation**): Per your excellent suggestion, we conducted comparative and ablation studies on a random sample of 31 recent CVEs (2025), which were reproduced and released by CVE-Genie. The results are summarized below:
> ||Success|
> |---|---|
> |Claude Code (Single-agent)|0|
> |CVE-Genie|48.4|
> |CVE-Factory|**90.3**|
> |*Ablation*||
> |Merge Stage 1-3|71.0|
> |Merge Stage 4-6|58.1|
>
>     As shown in the table, we employ Claude Code (SOTA Agent) as our single-agent baseline, which completely fails (0%) due to the complex, long-horizon nature of the task. In contrast, CVE-Factory (90.3%) significantly outperforms the existing multi-agent baseline, CVE-Genie (48.4%).
>
>     Furthermore, we explicitly validate our "decouple-couple" architecture through ablation studies. "Merge Stage 1-3" condenses the three decoupled generation stages into a single step to generate all files simultaneously. This overloads the agent's context, causing it to forget formatting constraints and generate progressively shorter files. "Merge Stage 4-6" collapses the step-by-step verification into a single end-to-end evaluation. This overwhelms the agent with simultaneous test-and-environment debugging, consistently leading to execution timeouts. These results demonstrate that both our granular decoupling and progressive coupling are strictly necessary for successful reproduction.

---

> > ### Author Rebuttal · Reviewer_d3Ky · 2026-04-01
> >
> > The rebuttal addresses some questions that I mentioned before. I will keep my positive attitude for this paper.

---

> > > ### Author Response · Authors · 2026-04-01
> > >
> > > Thank you for acknowledging that your concerns have been adequately addressed! We are grateful for the constructive feedback that helped improve our paper.
> > >
> > > We noticed that you selected "Fully resolved" and mentioned keeping a "positive attitude," yet the score remains unchanged. We respectfully want to understand: since the issues you raised have been resolved, could you kindly reconsider whether the current score still reflects your updated assessment of the paper?
> > >
> > > We believe the revisions have meaningfully strengthened the work, and we would greatly appreciate it if the score could be adjusted to align with the resolution of your concerns — as the acknowledgment option itself suggests ("please consider adjusting your score accordingly").
> > >
> > > Of course, **if there are any remaining reservations that we have not yet addressed, we would be happy to provide further clarification!**
> > >
> > > Thank you again for your time and effort in reviewing our work!

---

### Decision · Program_Chairs · 2026-04-30

**Decision:**

Accept (spotlight)

**Comment:**

This paper presents CVE-Factory, a multi-agent framework that automatically converts sparse CVE metadata into fully executable vulnerability repair tasks (environments, tests, and solutions) using a six-stage "decouple-couple" pipeline, and achieve significant performance improvements. All the four reviewers provided positive scores (4/4/5/5). Therefore, AC is to recommend acceptance.